# Dominance rank but not body size influences female reproductive success in mountain gorillas

Edward Wright[1]*, Jordi Galbany[2,3], Shannon C. McFarlin[2], Eric Ndayishimiye[4], Tara S. Stoinski[4], Martha M. Robbins[1]

**1** Max Planck Institute for Evolutionary Anthropology, Leipzig, Germany, **2** Department of Anthropology, Center for the Advanced Study of Human Paleobiology, The George Washington University, Washington, District of Columbia, United States of America, **3** Department of Clinical Psychology and Psychobiology, University of Barcelona, Barcelona, Spain, **4** Dian Fossey Gorilla Fund International, Atlanta, Georgia, United States of America

* edward_wright@eva.mpg.de

**Data Availability Statement:** All relevant data are within the manuscript and its Supporting Information files.

## Abstract

According to life history theory, natural selection has shaped trade-offs for allocating energy among growth, reproduction and maintenance to maximize individual fitness. In social mammals body size and dominance rank are two key variables believed to influence female reproductive success. However, few studies have examined these variables together, particularly in long-lived species. Previous studies found that female dominance rank correlates with reproductive success in mountain gorillas (G*orilla beringei beringei*), which is surprising given they have weak dominance relationships and experience seemingly low levels of feeding competition. It is not currently known whether this relationship is primarily driven by a positive correlation between rank and body size. We used the non-invasive parallel laser method to measure two body size variables (back breadth and body length) of 34 wild adult female mountain gorillas, together with long-term dominance and demography data to investigate the interrelationships among body size, dominance rank and two measures of female reproductive success (inter-birth interval $N = 29$ and infant mortality $N = 64$). Using linear mixed models, we found no support for body size to be significantly correlated with dominance rank or female reproductive success. Higher-ranking females had significantly shorter inter-birth intervals than lower-ranking ones, but dominance rank was not significantly correlated with infant mortality. Our results suggest that female dominance rank is primarily determined by factors other than linear body dimensions and that high rank provides benefits even in species with weak dominance relationships and abundant year-round food resources. Future studies should focus on the mechanisms behind heterogeneity in female body size in relation to trade-offs in allocating energy to growth, maintenance and lifetime reproductive success.

**Funding:** This research was funded by the Max Planck Society, National Geographic Society, The Columbian College of the George Washington University, The Wenner Gren Foundation (ICRG-123), The National Science Foundation (BCS 1520221) and The Leakey foundation.

**Competing interests:** The authors have declared that no competing interests exist.

## Introduction

Reproductive success is contingent on the appropriate allocation of energy among growth, reproduction and maintenance [1,2]. Due to the high energetic demands of gestation and lactation, this energetic trade-off is particularly critical in female mammals [3]. Female reproductive success can also be influenced by body size, body mass, body condition and health [4–7]. Furthermore, in group-living mammals, female reproductive success may be correlated with dominance rank and the strength of social bonds [5,8–12]. However, because of the difficulties in measuring body size in wild mammals, surprisingly few studies have examined the relationship between body size and dominance rank or correlated these variables to reproductive success [6,13].

According to life history theory large body size is predicted to be associated with costs and benefits. Allocating energy to attain and maintain large body size is associated with delays in reproduction, as energy allocated into one fitness-enhancing function cannot be concurrently invested into another [1,2,14]. Moreover, larger females have greater absolute metabolic needs than smaller ones, which can lead to increased risk of starvation during periods of resource unpredictability [15,16]. However, whilst trade-offs are evident in some populations, variation in individual quality can mask trade-offs in others [13,17,18]. Indeed, larger individuals often have higher fitness than smaller ones [19]. The proximate mechanisms behind this may include larger females having lower infant mortality than smaller ones, owing to larger mothers producing larger offspring, more or better quality milk or providing improved offspring protection [13,15,20,21]. Moreover, large female size may be associated with advantages in resource competition, as bigger females can outcompete smaller ones and are often dominant over them [15,16].

Factors other than body size may influence reproductive success. In many group-living species, high dominance rank is thought to confer females with priority of access to resources such as food [22,23], leading to better energetic condition [24,25] and higher reproductive success [10,26–28]. High ranking mothers in better energetic condition are able to invest more energy into reproduction, resulting in accelerated offspring growth [29,30], shorter inter-birth intervals [31–34] and lower infant mortality [24,31,35–37] than lower-ranking mothers. However, rank-related variation in energetic condition and reproductive success are only expected under certain ecological and social conditions [38–40].

One explanation for if and how ecological and social conditions can influence reproductive success is the socioecological model [28,38–40]. Socioecological theory posits that the degree to which resources can be defended by one or a few individuals influences the type of feeding competition and the strength of female dominance relationships, which in turn predict whether rank-related variation in energetic condition and reproductive success is expected [28]. In species living in environments where high quality resources are spatially or temporally clumped, leading to within-group contest competition and highly differentiated dominance relationships, higher-ranking females are expected to outcompete lower-ranking ones. Conversely, in species that feed on low value, evenly distributed food resources, dominance relationships tend to be weak and undifferentiated such that energetic condition and reproductive success are not expected to vary with individual dominance rank. However, socioecological predictions have been the topic of much debate, and few studies have rigorously tested the model predictions, particularly the relationship among dominance rank, energetic condition and reproductive success in the wild [28,40,41]. Moreover, species with weak dominance relationships have also been shown to have rank-related variation in reproductive success [36].

Female dominance rank is expected to correlate with body size (in addition to energetic condition) since body size typically correlates with fighting ability which commonly

determines dominance rank [10,42]. For example, larger female elephants, feral ponies and red deer are higher-ranking than smaller ones [43–45]. However, a number of traits, other than body size, have also been shown to be important in determining female dominance rank in group-living species, such as age, body mass and body condition [22,46–49]. In addition, in some species with female philopatry such as macaques, baboons and spotted hyenas, females occupy rank positions just below their mothers due to coalitionary support from kin, and consequently individual traits are not expected to correlate with dominance rank [31, but see 50].

Mountain gorillas are an interesting species to examine the interrelationships among body size, dominance rank and reproductive success. Gorillas are the largest extant primate and have one of the highest degrees of male biased sexual size dimorphism in mammals [51]. Female mountain gorillas have weak dominance relationships, which is expected for a species living in an environment with year-round abundant, evenly-distributed herbaceous vegetation [52,53]. However, these dominance relationships are stable over the long-term [53], and the majority of aggression is over food resources [54,55], suggesting that such dominance relationships may confer some benefits to high ranking individuals. Higher-ranking females have priority of access to some food resources and may have reduced energy expenditure compared to lower-ranking females due to decreased time travelling [55,56]. However, support for a positive correlation between dominance rank and energy balance (energy intake minus energy expenditure) was found in one population (Bwindi) but not in the other (Virunga population) [55–57]. Most interestingly, higher-ranking females in the Virunga population had significantly shorter inter-birth intervals and indications of lower infant mortality than lower-ranking ones (when each mother was a data point, but not when each infant was used as a data point) [58,59]. Given the low levels of feeding competition and weak dominance relationships, such relationships were not expected and those authors suggested that the positive correlation between dominance rank and reproductive success may in fact be a by-product of a positive correlation between rank and body size, such that body size is driving the relationship, not rank. However, it is unknown whether body size, which usually indicates fighting ability, is a strong correlate of rank in this species.

In this study we examined the interrelationships among adult female body size, dominance rank, and two measures of reproductive success, inter-birth interval and infant mortality, in wild mountain gorillas. Using the non-invasive parallel laser method [60–64] we measured two linear body dimensions associated with body size, back breadth and body length. We then tested the hypothesis that both morphological traits positively correlate with female dominance rank. Next, we tested the hypothesis that either higher-ranking and/or larger females had shorter inter-birth intervals and lower infant mortality than lower-ranking/smaller ones.

## Materials and methods

### Study population and photogrammetry

The study was conducted on 34 adult females monitored by the Dian Fossey Gorilla Fund's Karisoke Research Center, Volcanoes National Park, Rwanda (between 1˚21'and 1˚35'S and 29˚22 and 29˚44'E). We collected body size measurements between January 2014 and July 2016 using the non-invasive parallel laser method [60–64]. We measured two linear body dimensions, back breadth and body length as described in Wright et al. [64] and Galbany et al. [62,63]. These two measures incorporate several components of body size such as skeletal dimensions, and overlying tissue, including the rounded contours of the deltoid and gluteal muscles. Whilst static linear body measurements may not directly represent body condition or mass, which likely vary over time, they are expected to correlate with these measures and are important variables potentially influencing reproductive success [6,21,65]. Females attain 98% of their body length and

back breadth by 11.7 and 11.9 years, respectively [63]. Therefore, we only measured females aged 12 years and above. Photographs were collected and measured in ImageJ [66] by E.W. and J.G. [see S1 Material and [64] for details of photogrammetry error]. Measurements were obtained from an average of six photographs per female and trait (range: 3–10) totaling 420 photographs. Back breadth and body length were weakly positively correlated ($r_s = 0.38$).

## Dominance hierarchies

Dominance hierarchies were based on displacement and avoids (approach and retreat interactions) collected since the formation of each group or since 2000 (whichever was earliest) until July 2016, during focal animal follows and ad libitum observations [53,67,68]. Approximately four hours of observation were made on each group on a near daily basis. Females were considered behaviourally mature from age 8 years and older [69], so we included dominance data on all females age eight and above (regardless of whether we had their body size measurements). Dominance hierarchies were computed using the Elo rating method [70,71]. Females were given a starting value of 1000 and k was set to 100. Maturing and immigrating females entering into the dominance hierarchy were set to the lowest Elo rating of that day (Elo rating argument innit set to bottom). This was based on indications from previous studies that immigrating females receive higher rates of aggression upon immigrating into a group from resident females and that dominance rank is in part related to tenure duration in the group [53,72]. Additionally, we employed a burn in period, and only considered female dominance ranks to be accurate once a female had interacted for a minimum of ten times. Ranks were standardized per group and day such that the lowest ranking female was assigned 0 and the highest ranking female 1 and rankings in between were set proportionally to their Elo rating. The mean number of dominance interactions per female was 62 during an average of 10.4 years per female (range: 13–165; SD = 33; Fig 1).

## Inter-birth intervals

Inter-birth interval was defined as the interval between two successive births by the same mother. We only considered inter-birth intervals in which the first offspring in the interval survived to weaning age (three years), as previous studies have shown that inter-birth intervals are shorter following the death of unweaned infants [59,73]. This ensured that the observed variance in inter-birth interval was not driven by infant mortality. The analysis included infants born between April 2000 and April 2014 from 16 females. Demography data was used until April 2017 to determine the survival of the first infant in all inter-birth intervals. All births were known to the nearest day.

## Infant mortality

We included 64 infants born between April 2000 and July 2015 from 28 females. We recorded whether each infant survived to weaning age, using demography data up until July 2018. We excluded cases of infanticide ($N = 6$) from the analysis because we were focusing on variation in mortality related to body size and/or dominance rank. Infanticide by males occurred following group disintegrations and interactions with lone silverbacks and neither female dominance rank or body size were likely to prevent it.

## Statistical analyses

To test the hypothesis that back breadth and body length were positively correlated with dominance rank, we fitted a beta model with logit link implemented with the r function

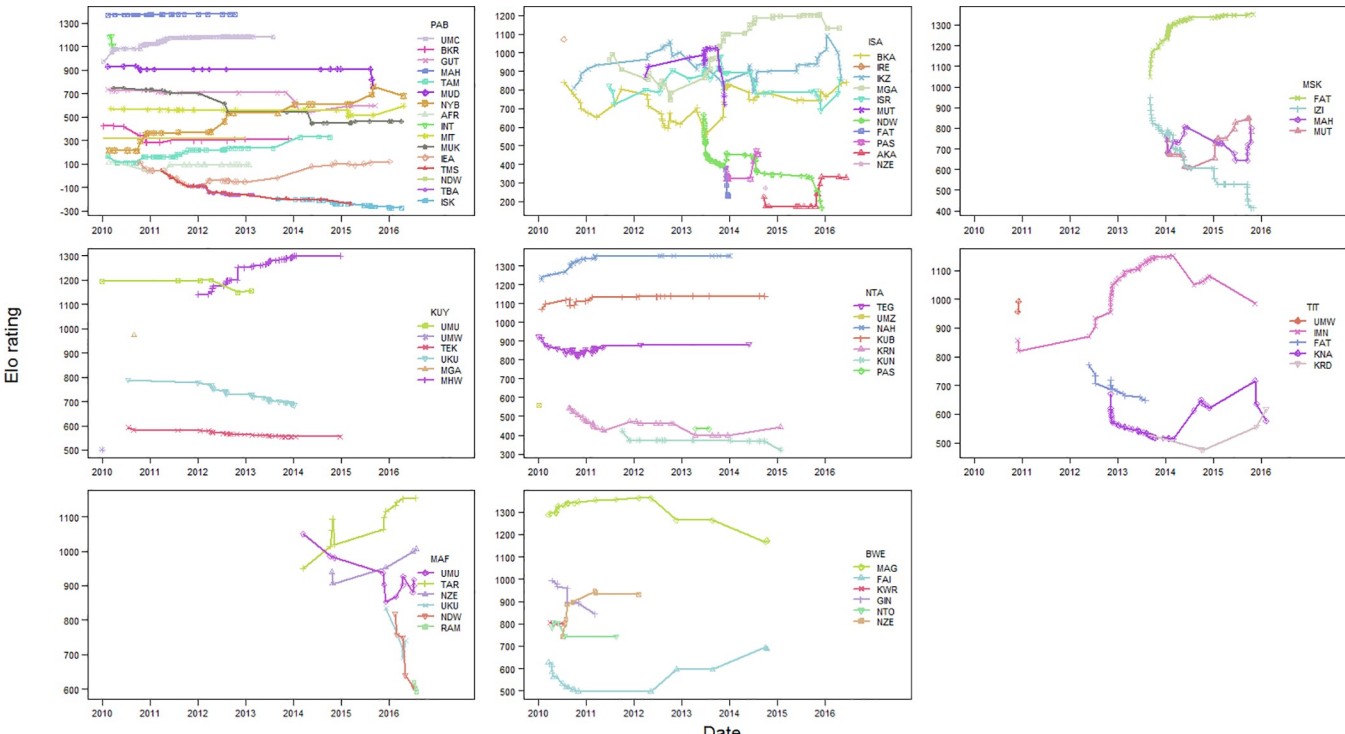

**Fig 1. Elo ratings for adult females in eight social groups between January 2010 and July 2016.** Elo ratings were calculated using long-term dominance interactions dating back to the date of each group formation (or to 2000 for PAB group). For clarity data are only shown for the period 2010–2016. Each symbol denotes a female and each point represents an interaction between females, with lines between points representing the change in Elo rating for the two interacting females.

"glmmTMB" of the "glmmTMB" package [74]. The response variable was dominance rank (averaged over the photogrammetry period: January 2014 –July 2016 and standardized between 0 and 1, see above). Each female was a data point (*N* = 34). We included back breadth and body length as test predictors, age (averaged over the photogrammetry period) as a control variable and group ID (at the time of photogrammetry) as a random effect. We also included the random slopes of back breadth and body length within group ID [75]. Because our hypotheses concerning the effects of back breadth and body length on dominance rank were based on the relative difference between females within a group rather than absolute differences across groups we centered back breadth and body length among females within each group to a mean of zero [64] (the largest difference in the group mean across groups was 3.4 cm and 4.4 cm for back breadth and for body length, respectively).

To test whether dominance rank, back breadth or body length significantly influenced inter-birth interval duration we fitted linear mixed models (LMMs) with Gaussian error distribution and identity link implemented with the r function "lmer" of the "lme4" package [76]. The response variable in each model was the inter-birth interval and each inter-birth interval represented a separate data point (*N data points* = 29; *N females* = 16). We fitted three models, one for each predictor: dominance rank, back breadth and body length (due to the low number of data points in the model; we also fitted a multivariate model comprising all the predictors, see below). Values for dominance rank were taken from when the interval started (birth of the first infant in the inter-birth interval). We included group ID and mother ID as random effects in each model. Ideally, we would fit one model with all three predictors as well as controlling for the potential influence of several additional variables such as mother age, offspring sex (12

F; 18 M), parity of mother (4 primiparous; 25 multiparous) and whether the mother transferred in between the birth of the two infants (23 did not; 6 transferred) [59,73,77,78] but due to the low number of inter-birth intervals this likely results in an over-parameterized model with very low power. Nevertheless, a potentially over-parameterized model with all test and control predictors can still be informative in indicating which variables are likely important in influencing inter-birth interval duration (this model is presented in the S1 Material).

To examine whether dominance rank, back breadth or body length were significantly associated with the likelihood that infants survived to weaning age we used mixed effects Cox proportional hazards models [79] using the "coxme" function from the package coxme [80]. The response variable in each model indicated for each infant (the data point; *N data points* = 64; *N females* = 28) the number of days which had passed until weaning or at death (if the infant died before weaning age), and the status variable indicated whether the infant was alive (0) or dead (1) at the time of weaning. We fitted three separate models, one for each test predictor: dominance rank, back breadth and body length (due to the low number of data points in the model; we also fitted a multivariate model comprising all the predictors, see below). We included group ID and mother ID as random effects in each model. As above we also fitted an additional and potentially over-parameterized model with the three test predictors and three additional control variables: mother age, group size and parity of the mother (8 primiparous; 56 multiparous; see S1 Material for further information on this model).

All the analyses were conducted in R [81]. We checked for overdispersion in the glmmTMB model. The dispersion parameter was 1.2, close to the ideal value of 1 [82]. We also checked for normally distributed and homogenous residuals of the LMMs by visually inspecting qq-plots and residuals plotted against fitted values. In the model with multiple predictor variables we checked for collinearity among the predictors by examining variance inflation factors using the "vif" function from the "car" package [82]. In addition, we checked for model stability by re-running the models after excluding each level of the random effects one at a time and comparing the estimates derived from these models with the estimates from the original model on the full data set. No stability issues were found. All quantitative predictors were z-transformed in each analysis (to a mean of 0 and standard deviation of 1). Before determining the significance of individual predictors we compared a full model with a corresponding null model [excluding the test predictors of interest; 83] using likelihood ratio tests. P-values for individual predictors were also derived through likelihood ratio tests, comparing a full model with a reduced model not comprising the test variable (excluded one at a time). Confidence intervals were determined using the functions "simulate.glmmTMB" and "bootMer" of the "glmmTMB" and "lme4" packages.

### Ethical note

The Rwanda Development Board and the Ministry of Education gave permission to conduct this study.

## Results

The mean back breadth and body length were 48.9 cm and 71.1 cm, respectively (Table 1; for reference the corresponding male values are also displayed [64]).

Back breadth and body length showed no significant associations with dominance rank (non-significant comparison of the full model comprising back breadth and body length with a null model with these variables excluded: likelihood ratio test: $X^2$ = 2.03, df = 2, $p$ = 0.363).

The mean inter-birth interval was 44.5 months (*N* = 29; *SD* = 6.6; range = 34.7–63.1). Higher-ranking females had significantly shorter inter-birth intervals than lower-ranking ones

**Table 1. Mean, range, sample size (*N*) and coefficients of variation (CV) among females and males for the two morphological traits: Back breadth and body length.**

| Trait | Back breadth | | Body length | |
|---|---|---|---|---|
| Sex | Female | Male [†] | Female | Male [†] |
| Mean | 48.9 | 59.2 | 71.1 | 87.7 |
| Range | 45.0–52.1 | 54.6–65.0 | 64.2–78.7 | 80.8–96.5 |
| *N* | 34 | 26 | 34 | 26 |
| CV % | 3.6 | 4.9 | 4.0 | 3.3 |

[†] Taken from Wright et al. [64]. CV % is calculated by dividing the standard deviation by the mean, multiplied by 100. For intra-individual CV % see S1 Table in S1 Material.

(LMM, estimate ± *SE*: -2.570 ± 1.209, 2.5 and 97.5% confidence intervals: -5.166; -0.170, $X^2$ = 4.201, *df* = 1, *p* = 0.040; Fig 2). An increase in one standard deviation in dominance rank resulted in a reduction of 2.6 months in inter-birth interval. Neither back breadth nor body length significantly influenced inter-birth interval duration (LMM back breadth, estimate ± *SE*: -0.230 ± 1.381, 2.5 and 97.5% confidence intervals: -3.191; 2.794, $X^2$ = 0.026, *df* = 1, *p* = 0.871; LMM body length, estimate ± *SE*: 0.844 ± 1.361, 2.5 and 97.5% confidence intervals: -3.048; 2.810, $X^2$ = 0.382, *df* = 1, *p* = 0.537, respectively; the model including the three test predictors and all control variables revealed similar results, see S1 Material).

Of the total 64 infants that could have reached age 3 years during the study, 23 (35.9%) did not survive to weaning age. Infant mortality was not significantly correlated with dominance rank, back breadth or body length (Cox LMM dominance rank, estimate ± *SE*: 0.260 ± 1.297, $X^2$ = 1.191, *df* = 1, *p* = 0.275; Cox LMM back breadth, estimate ± *SE*: -0.255 ± 0.274, $X^2$ = 0.932, *df* = 1, *p* = 0.334; Cox LMM body length, estimate ± *SE*: 0.133 ± 0.270, $X^2$ = 0.299, *df* = 1, *p* = 0.585; the model including the three predictors and all control variables revealed similar results, see S1 Material).

## Discussion

We examined the interrelationships among body size, dominance rank and reproductive success in female mountain gorillas. Neither linear body dimension was clearly associated with variation in dominance rank or reproductive success. However, higher-ranking females had significantly shorter inter-birth intervals than lower-ranking ones, confirming results of earlier studies [58,59]. Assuming dominance rank is related to access to resources as suggested by other studies [55,56], our results suggest that rank may influence female body condition, which may be a stronger correlate of reproductive success than linear body size in this species. Even though our linear measures of body size incorporate some components of muscle size, we cannot rule out that other measures such as body mass or body condition would reveal different relationships.

### Body size and dominance rank

We found no clear support for back breadth or body length significantly correlating with female dominance rank. This finding went against our expectation as body size tends to correlate with fighting ability, or levels of aggression, which commonly determine dominance rank [42,84]. This result contrasts with some other group-living species such as female feral ponies, elephants and red deer [43–45]. The lack of a strong correlation between linear measures of body size and dominance rank is also surprising as positive relationships between body mass and female dominance rank appear to be common [22,47–49,85]. However, in some species,

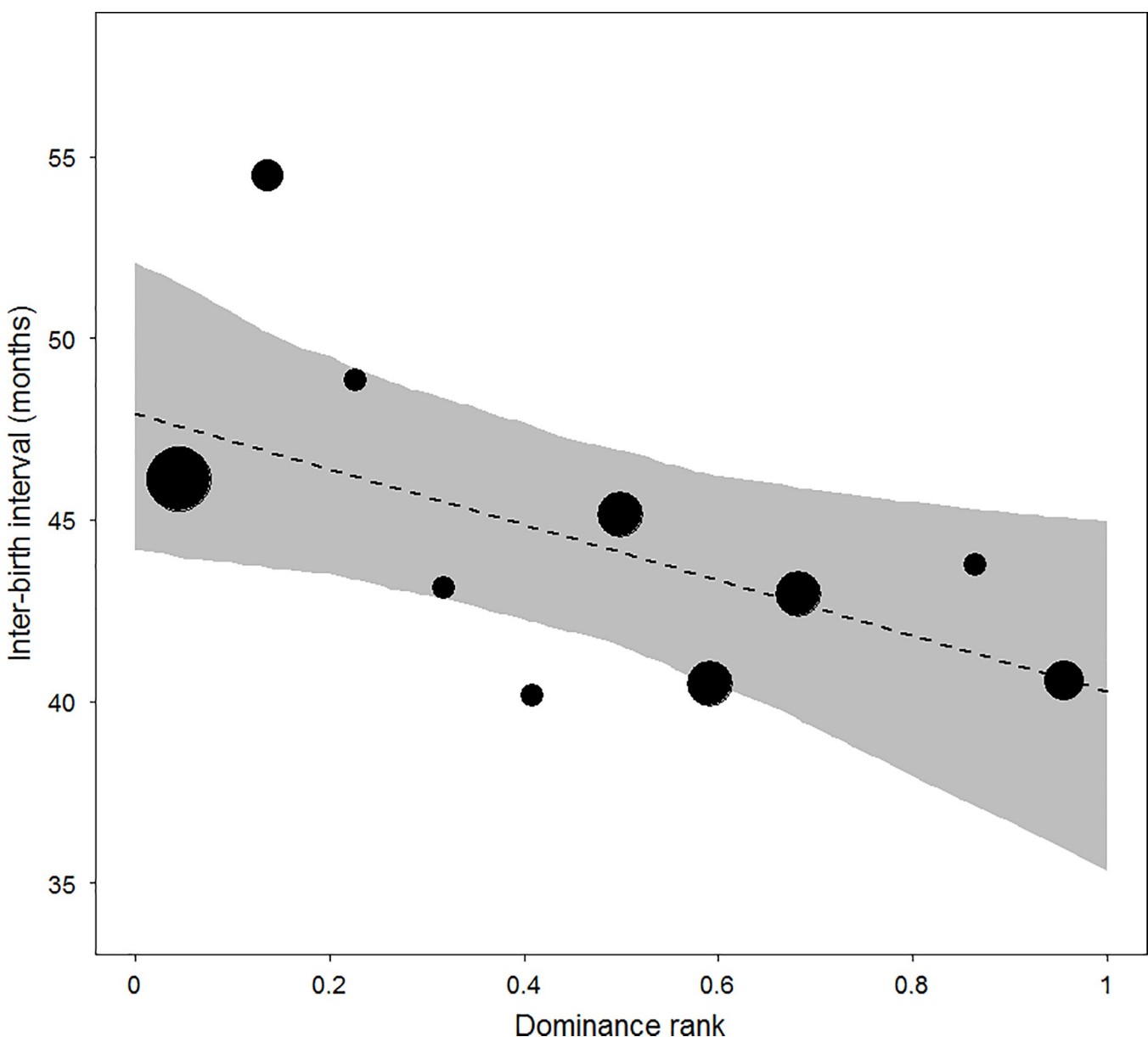

**Fig 2. Relationship between dominance rank and inter-birth interval duration.** Dominance rank is standardized between 0 (lowest rank) and 1 (highest rank). The area of the circles represents sample size ($N = 29$ inter-birth intervals; 16 adult females). The dashed line is the fitted model line and the shaded area is its 95% confidence intervals.

high dominance rank may lead to increases in body mass, rather than being a consequence of large body mass [48]. In female mountain gorillas, traits such as age and group tenure, which have been shown to influence dominance rank, are likely to have a greater importance in determining dominance rank than body size [53]. Moreover, these results suggest that female dominance rank may not strongly reflect (current) fighting ability or that larger females do not often challenge smaller higher-ranking ones. Female gorillas exhibit low rates of aggression towards each other, particularly aggression involving fighting or physical contact [52,54]. These results are similar to those for female chimpanzees, which do not tend to aggressively challenge rank positions and have stable long term dominance relationships that are, at least in part, dependent on group tenure length [86].

The lack of a significant relationship between back breadth and female dominance rank contrasts with findings in male mountain gorillas in this population, which show strong positive correlations between back breadth and dominance rank [64]. Since male fitness is more dependent on access to mates, which are a more limited resource than food, large body size and associated advantages in fighting ability are expected to be under much stronger selection in males [87]. Therefore, compared to males, females may follow a more conservative growth strategy, allocating resources into reproduction and maintenance rather than large body size. In support for this idea, females attain full body size at an earlier age than males [63,88] and have their first offspring 4–5 years earlier on average than males [89]. Variation in body size among females would also be expected to be lower. Accordingly, we found that the variation (coefficients of variation; CV) in back breadth among females was smaller than it was among males (Table 1). Interestingly, variation in female body length was slightly higher than the variation in male body length, although it is unclear why this is the case (Table 1). We also found no significant associations between body length and either female or male dominance rank [current study, 64], which suggests that this trait does not correlate with competitive ability in either sex.

## Body size, dominance rank and reproductive success

We found that higher-ranking females had significantly shorter inter-birth intervals than lower-ranking ones, which is similar to findings in earlier studies on this population [58,59], and to other group-living species [31,32,34,90,91]. The main explanation for this relationship is that high dominance rank leads to priority of access to food resources and consequently better energetic condition [10,26,27,32]. Evidence for rank-related variation in energetic condition has been found in the neighboring Bwindi mountain gorilla population [56]. Higher-ranking females had higher energy intake rates, due to faster ingestion rates, as well as lower energy expenditure than lower-ranking females, leading to a positive relationship between dominance rank and energy balance. However, support for rank-related energetic condition in the current population has received less support [55,57]. Even though higher-ranking females had greater access to some food resources over lower-ranking ones, dominance rank did not significantly predict energy intake rate or levels of urinary C-peptide, a common proxy for energy balance. To further investigate the mechanism leading to shorter inter-birth intervals in higher-ranking females, future studies should measure the phases of the inter-birth interval separately (estrous cycling, gestation and lactation). In baboons, higher-ranking females have shorter postpartum amenorrhea (lactation) phases than lower-ranking ones [34], whereas in mandrills it is the cycling phase that is reduced in higher-ranking females [33]. More generally, high dominance rank may also confer other benefits to high ranking individuals, which were not investigated here, such as improved mate choice, lower predation risk and reduced social stress [10].

We found no support for linear body dimensions influencing inter-birth intervals. This suggests that large female size in mountain gorillas does not provide clear advantages in resource competition that could result in improved energetic condition, assuming that additional energy accrued via resource competition is allocated to reproduction and not only maintenance of larger body size. This result contrasts with fur seals, for example, in which females with larger linear body dimensions are in better energetic condition and therefore invest more energy into offspring than smaller ones, leading to improved reproductive success [13,92]. In general, larger females are able to store greater fat reserves than smaller ones, allowing them to invest more resources into offspring [7,92,93]. However, the advantage of storing greater fat reserves is only expected to benefit capital breeders, such as harbor and fur seals as well as

many other large mammals, which rely on stored energy reserves to meet the higher costs of gestation and lactation [94]. In contrast, strong selection for large body size in order to increase storage of fat reserves is not expected in income breeders which rely on short-term food acquisition to meet the increased energy demands of reproduction [1,7]. It is unclear where mountain gorillas lie on the capital-to-income breeder continuum, although during periods of increased energetic need, we would expect them to rely on fat reserves to some extent, as observed in orangutans *Pongo abelii* [95]. However, mountain gorillas live in an environment with year-round abundant food resources. Therefore, selection for large body size to better store fat reserves for use during costly reproductive phases may be reduced compared to other large mammals. Interestingly, body mass correlates with dominance rank and reproductive success in female chimpanzees [47,91]. Overall, we suggest that body condition, which is likely influenced by dominance rank, is more important in determining female reproductive success than body size in mountain gorillas.

We did not find support for linear body dimensions or dominance rank influencing infant mortality, similar to a previous study [59]. This contrasts with several other studies which found that high dominance rank and/or large female size to be generally associated with lower infant mortality in red deer, long-tailed macaques, spotted hyenas and chimpanzees for example [4,24,31,36], although this relationship is not universal [10,27]. The influence of body size and dominance rank on infant mortality is likely to depend on other causes of infant mortality including species specific predation pressure, the risk of infanticide and the degree of feeding competition [96]. Lastly, in primates at least, it has been suggested that the advantages of high dominance rank may be stronger via its effects on infant growth and rates of reproduction rather than infant mortality [27].

Sexual dimorphism is the product of selection acting on both male and female body size [15,20]. Since we did not find clear support for advantages (or disadvantages) of large body size in females, as measured by two linear dimensions, we suggest that sexual dimorphism in mountain gorillas may be a product of selection on large male size, presumably due to strong male-male competition [64,88]. Furthermore, the large size of female gorillas can be explained by genetic correlation between the sexes for genes controlling for size [15,97].

## Conclusion

Females must balance the need to begin reproducing as early as possible against attaining sufficient body size to optimize reproduction across the lifespan and maximizing survivorship. An interesting question is whether body size correlates with longevity, because higher-ranking female mountain gorillas live longer than lower-ranking ones and consequently they produce more surviving offspring over the lifespan, resulting in higher lifetime reproductive success [98]. In addition, studies should examine the influence of maternal effects and early life adversity on heterogeneity in adult female body size, longevity and lifetime reproductive success [99,100]. Correlating body size with lifetime reproductive success would be informative about the life history trade-offs of maintenance, reproduction and survival, but such data are difficult to obtain [4,36].

## Supporting information

**S1 Material.**
(DOCX)

**S1 Table. Body size and dominance rank analysis.**
(XLSX)

**S2 Table. Inter-birth interval analysis.**
(XLSX)

**S3 Table. Infant mortality analysis.**
(XLSX)

## Acknowledgments

We thank the Rwanda Development Board and the Ministry of Education for permission to conduct research in Volcanoes National Park. We are indebted to all current and previous staff of the Dian Fossey Gorilla Fund International's Karisoke Research Center for their dedication in the monitoring and protection of the gorillas. We thank Winnie Eckardt and Andrew Robbins for helpful discussion on the project as well as Felix Ndagijimana and Veronica Vecellio for support. We also thank Roger Mundry for statistical advice and the two reviewers for helpful insights which improved the manuscript.

## Author Contributions

**Conceptualization:** Edward Wright, Martha M. Robbins.

**Data curation:** Edward Wright.

**Formal analysis:** Edward Wright.

**Funding acquisition:** Shannon C. McFarlin, Martha M. Robbins.

**Investigation:** Edward Wright, Jordi Galbany, Eric Ndayishimiye.

**Project administration:** Tara S. Stoinski, Martha M. Robbins.

**Supervision:** Martha M. Robbins.

**Writing – original draft:** Edward Wright, Martha M. Robbins.

**Writing – review & editing:** Edward Wright, Jordi Galbany, Shannon C. McFarlin, Tara S. Stoinski, Martha M. Robbins.

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
