## [Decision Letter · Decision Letter 0]

2 Dec 2019

PONE-D-19-28202

Dominance rank but not body size influences female reproductive success in mountain gorillas

PLOS ONE

Dear Dr. Wright,

Thank you for submitting your manuscript to PLOS ONE. After careful consideration, we feel that it has merit but does not fully meet PLOS ONE’s publication criteria as it currently stands. Therefore, we invite you to submit a revised version of the manuscript that addresses the points raised during the review process.

Both reviews appreciated the qualty of the work but also highlighted some important concerns. I would recommend to carefully consider the comments in your revision and specifically address the reviewers concerns about the statistical approach and its justification.

We would appreciate receiving your revised manuscript by Jan 16 2020 11:59PM. To enhance the reproducibility of your results, we recommend that if applicable you deposit your laboratory protocols in protocols.io, where a protocol can be assigned its own identifier (DOI) such that it can be cited independently in the future. For instructions see: http://journals.plos.org/plosone/s/submission-guidelines#loc-laboratory-protocols

We look forward to receiving your revised manuscript.

Kind regards,

Julien Martin

Academic Editor

PLOS ONE

Journal Requirements:

Reviewers' comments:

Reviewer's Responses to Questions

**Comments to the Author**

1. Is the manuscript technically sound, and do the data support the conclusions?

Reviewer #1: Yes

Reviewer #2: Yes

2. Has the statistical analysis been performed appropriately and rigorously? 

Reviewer #1: No

Reviewer #2: Yes

3. Have the authors made all data underlying the findings in their manuscript fully available?

Reviewer #1: Yes

Reviewer #2: Yes

4. Is the manuscript presented in an intelligible fashion and written in standard English?

Reviewer #1: Yes

Reviewer #2: Yes

5. Review Comments to the Author

Reviewer #1: This study is one of the first to identify the relationship between dominance rank, body size, and reproductive success in a wild mammal – an important question in the context of life history evolution. The authors collected two different body size measures from 34 wild female gorillas via parallel-laser photogrammetry, a method that they have used in several other publications. They hypothesize that (1) body size and dominance rank are positively associated, and (2) larger size and/or higher rank confer reproductive benefits such as shorter inter-birth intervals and higher infant survival rates.

This is an interesting and generally valid study. The authors found an association between higher rank and shorter inter-birth interval; however, they reported no association between body size and (1) dominance rank, (2) inter-birth interval, or (3) infant survival. I find the negative findings as helpful as the positive ones, and I am glad they include these three different analyses.

Most of my suggestions are minor, and most simply serve to make the manuscript clearer for the reader. However, I have suggested a few more major changes that I think are necessary for publication and will greatly improve the introduction and data analysis. I think this study helps to fill in large gaps in our understanding of wild animal body size, especially in females, and its relationship to other demographic and life-history information.

Thanks for providing an exciting read on an important topic.

Best,

Emily Levy

Major:

1. Introduction/framework

The introduction does a great job giving some relevant background information – which is a fair amount, as you must introduce literature on dominance rank, body size, reproductive success, and relationships between all of these factors. But, I think it would benefit from some re-organization and condensing. Currently there are a lot of paragraphs, each seeming to talk about something somewhat unique, which leaves the reader a bit strung out and confused and waiting to see how (and in what species) it will all come together. One suggestion is to move most of the text about the socioecological model to the discussion section titled ‘body size and dominance rank,’ when you discuss that female gorillas show low rates of aggression.

Second, the study is framed as a test of a trade-off between adult body size and reproductive success, but such a trade-off isn’t as well-supported in the literature as the introduction makes it seem. I know of no evidence of such a trade-off in within-species studies of primates. The second paragraph discusses this trade-off, but I think most of the citations are of between-species comparative studies of primates. While these studies are helpful, patterns between species do not necessarily follow within-species (e.g., link between size & lifespan across species vs. within domestic dogs).

This idea of a trade-off is presented in the introduction and discussion. While your limited amount of energy intake must be allocated to growth, reproduction, and/or maintenance, things are a little more complicated when you consider that most growth happens before reproduction (especially in this study, where you measure females that have finished growing). So, as infants/juveniles, individuals could allocate energy to skeletal growth or maintenance but not reproduction; as adults, individuals could allocate energy to muscle mass, reproduction, or maintenance. In addition, differences in early-life environmental conditions across females in your study may also affect the extent to which they experienced any trade-offs. This is further complicated by the fact that variation in female quality may mean that some females can ‘have it all’ while others are forced to make allocation decisions. Finally, many within-species studies demonstrate that individuals who are larger also have increased fitness components (Kingsolver & Pfennig 2004 is a good meta-analysis of that). So, I think the study’s framing should be shifted away from testing for a trade-off.

2. Data analysis

For each gorilla, you take a mean body size and use that as the unit of analysis in the models. While this is common in the photogrammetry literature, using each photograph as the unit of analysis (and adding a random effect of individual) provides the model with within-individual variance. You could imagine a scenario in which you only have 3 photos of the two largest individuals and 10 photos of everyone else. The fewer photos you have of an individual, the smaller the probability that your mean is the true measurement of that individual. As is, your models will give equal weight to those two large individuals as everyone else. In addition, your models will not account for the fact that variance in some individual’s photos are a lot larger than in other individual’s photos. Ie, you’re more confident in the mean when SD=0.1 than when SD=1, all else being equal. I don't think you need to change the analysis for publication, but if not, you should justify your current approach.

There is no reporting on whether (and to what extent) the photogrammetry data enable you to differentiate between individuals. In other words, is the ratio of intra-individual error : inter-individual variance low enough for you to detect an effect of body size in the first place? One way to ask this is to ask what proportion of the variance in your body size measures are explained by individual identity. Another is to do a power analysis showing that given your sample size and the error in your photographs, you would be able to detect an effect size XX large. At the very least, a table with error/variance and/or a scatterplot with a point for each individual and SD bars around it would help the reader visualize what the data look like. Finally, it would be beneficial to indicate whether body size differed between groups, which would explain why you centered the data for one of the analyses.

Minor:

On line 64, you say that “individuals should aim to find a balance among the benefits and costs of attaining optimal body size to maximize reproductive success.” I think this statement would be more accurate if written from an evolutionary perspective – it currently seems to be about individuals finding that balance, when their ability to make those decisions has (probably) been shaped by selection.

On line 67, you say body mass is associated with reproductive success. The next sentence states that ‘it is not always clear which component of body mass is responsible for driving such relationships.’ It’s not clear to the reader why one over another (eg, a larger skeleton vs. more muscle mass) would matter or be interesting.

On line 163 you refer the reader to your recent male body size & dominance rank manuscript for data on photogrammetry error. While that’s helpful for a general photogrammetry overview, I think you should still include the error associated with the images used in the current study (ie, within-individual %CV; see ‘major’ comment above).

On line 189, you don’t specify if the time period for the mean number of dominance interactions per female. (e.g., 16 years? 1 year?)

Why did you switch between GLMMs and LMMs between the 1st and 2nd set of models? They seem extremely similar in construct, and glmmTMB allows several different inputs for data distributions. A brief explanation for the change would help.

On line 258 and 266, please provide citations when discussing dispersion and stability – I’m not familiar with what an ‘allowed’ cutoff is, and readers might not be either.

Your first model (rank ~ back breadth + body length) doesn’t include age – perhaps a citation that rank isn’t associated with age in female gorillas (or an additional model or correlation?) would justify this, as it’s common for dominance rank to change with age across both sexes of many species.

Your description of the model results were extremely clear, and I especially appreciated that you explained the ‘effect size’ of your model coefficients. One question: On line 280, I’m confused as to why you didn’t give a separate X2 and p-value for each body size measurement individually by removing one, running log likelihood, adding it back and removing the other.

On line 354, you state that back breadth %CV was ‘considerably smaller’ in females than males. However, %CV of length is larger in females than males, though the male-female difference is smaller for length than breadth. If you’re going to focus on breadth, I think you need to (1) acknowledge the length difference, and (2) run stats to show whether/that breadth %CV is appreciably larger. Also – is this the %CV reported here the average of each female’s %CV?

In line 379-380 it sounds like there’s “little support” high rank being associated with better energy condition – does this mean there’s no support, or there’s enough to tell us about? I think fleshing this out more will be helpful, especially because you discuss the findings from Bwindi. From line 381 to the rest of the paragraph, the two ideas brought up (rank*offspring sex and future studies of IBI phases) seem like non-sequiturs after the longer discussion of rank & resource access.

In lines 416 and 418, it would be helpful to list the species in the studies you cite.

Lines 426 – 432, which are the start of the conclusion, would work better as a separate paragraph in the discussion. The focus of the study (in the intro) was not sexual dimorphism in gorillas.

Reviewer #2: This interesting manuscript seeks to understand if the unexpected relationships between dominance and different measures of female reproductive success found in previous studies from multiple mountain gorilla populations could be linked to a correlation between body size and dominance rank. Body size is normally strongly correlated to dominance rank in multiple mammalian species, granting priority access to resources that can in turn be used for increased allocation to body growth, maintenance, and to reproduction. However, this works shows that there seems to be no correlation between linear body size measures and dominance as well as with multiple indexes of reproductive success in mountain gorilla females.

Although no effect was found, I think this study could be an interesting contribution to our knowledge on the determinants of dominance among females in sexually dimorphic species living in certain ecological and social conditions. I do have, however, some concerns about how the analyses were carried out and the absence of important statistical/methodological details is something that should be addressed (see below). If something is not clear or if I can be of further help please feel free to contact me (email: luca.montana@usherbrooke.ca).

Best wishes,

Luca Montana

Major comments

1. I understand the limitations imposed by the limited sample size on the way that statistical analyses were carried out, but I do think that it would be better if the ms compared biologically meaningful models, instead of presenting three univariate models and a full model. If multiple biologically meaningful hypotheses were translated into statisticasl models, they could be tested to understand which parameters affect female reproductive success, for example: dominance, body size, a combination of both, and possibly controlling for different parameters such as infant sex, parity etc. If this was done in model selection framework, the AIC estimated by the model selection would penalize overparametrized ones and should help understand which one is the best of the proposed models.

2. Related to the my comment above, this specifically applies to both inter-birth interval and infant survival predictions:

a. Why are back breadth and body length not joined in the same ‘body size’ model? Since both measures describe linear body size, I’m not sure I understand why they are not included in one multivariate model testing if body size has an effect on dominance, inter-birth interval duration, or infant survival. Using the rule of thumb described in Dormann et al. (2013) two variables correlated by rs < 0.7 can be used in the same model.

b. I think it would be interesting to compare the effects of dominance rank on inter-birth interval duration of the univariate model vs the full model. Although the full model is not significantly different from the null model, dominance is significant in both models. If the effect of dominance is different between the two models, this could mean that other parameters influence the effect of dominance on reproductive success and thus the univariate model it’s not the best way to make predictions.

3. I think the ms would benefit from more information about the choice of the models used to test the interesting hypotheses presented here. I specifically pointed out some examples in my minor comments

Minor comments

4. L. 67-68. This is really petty and I apologize to pointing it out, but the sentence says that body mass is positively correlated with female reproductive success in mammal, but then a study on both birds and mammals is cited. I did not go through the cited paper (Ronget et al. 2018), but is the correlation showed just on mammal or on both birds and mammals?

5. L. 73. I think something is either missing or the article ‘a’ should be removed from the sentence.

6. Ll 107-109. In its current state, the sentence argues that body size is correlated with female dominance only in social species. Wouldn’t that be the case also for group-living but not strictly-social species?

7. Ll 120-122. I admit I was surprised by the fact that gorillas had one of the highest degree of male biased ssd in mammal. I thought other mammalian species showed higher degrees than that (not true: Jarman (1983)), but I found out that many times it’s not the average that is reported (e.g. Rioux-Paquette et al. (2015): males can weight up to 4 times as much as the smallest breeding female). Today I learned multiple new things!

8. Ll. 126-129. In my opinion, this is a critical part of the introduction but I find it sometimes confusing. May I suggest to spend some more words to better introduce the correlation between dominance and priority of access to some food resources etc? Also, I might be wrong, but I think that it was Wright et al. (2014) that described the decreased travelling time? So reference 64 instead of 61.

9. Ll. 164-168. At first I started looking for the sample size here, and I realized only later that N changed depending on the analyses. Maybe it could be a good idea to specify it here? Also, related to N, since repeated measurements of the same individuals are present, I think it could be a good idea to specify both the total number of measurement and the number of females measured.

10. Ll. 171-189. I really appreciate the details on how dominance was estimated. Good job! However, related to the next comment and to comment n. 3, only by looking at Figure 2 I understood that ranking was treated as a continuous variables that ranges from 0 to 1. I think it would be appropriate to specify how this was done, because by looking at Figure 1 Elo-rating values range between -300 and about 1350. Also, was dominance rank standardized or scaled between 0 and 1? It’s not clear to me how the values I just mentioned could be standardized between 0 and 1.

11. L. 218. I did not understand how a beta distribution could be used to test if back breadth and body length were correlated with dominance rank until I saw that dominance ranks were a continuous variable ranging between 0 and 1.

12. Ll. 222-223. I think a small sentence explaining why the random slopes of back breadth and body length were used within group ID (which I think means ‘back breadth | group ID’ and ‘body length | group ID’). I can see why the effect of body measurements could change depending on the group, but I think readers could benefit from a short sentence on why these random slopes were used.

13. L. 227. Not sure about the following reference: van de Pol and Wright (2009). I think it could mislead people looking for papers using subject centering. I think the explanation in ll. 223-227 is really clear and I agree that this was the best way to see if morphological measures had an effect on dominance rank based on relative differences between females within the same group.

14. Ll. 239-240. Since I’m not working with a species that forms stable groups it took me a while to understand the “…the mother transferred in between the birth of two infants…” meant that females could move to different groups between giving births. I think it could benefit the readers to add a sentence when talking about the natural history on the species.

15. Ll. 247-250. I’m no expert of Cox proportional hazardous models, but aren’t they used to make prediction when using censored data? I do not understand if censored data are present (related to comment n. 3), and if not why Cox regression were the model of choice. Instead of a Cox regression, wouldn’t a binomial glmm have been enough to estimate the probability that a young reached weaning age?

16. Ll. 270-271. Given what is stated in ll 223-227 (centering of back breadth and body length among females within each group to a mean of zero), I assume that these two measurements were z-transformed too, am I right? This would mean that back breadth and body length were z-transformed differently depending on the hypothesis that was being tested. If this is the case, I think it could help readers if something was said about this matter either here or at ll 223-227.

17. Lines 320-323. Given the controversial results on different populations of the same species I think here there are lots of assumptions that are used to say that rank may influence body condition (or the other way around?), which may be in turn a stronger correlate of reproductive success than linear body size. The only way to test this would be through structural equation modelling (Shipley 2002). Similar comments below.

18. Ll. 329-332. There are lots of contradictory studies on the correlation dominance rank-body size in females (Clutton-Brock and Huchard 2013). It appears to me that body size was shown to correlate with dominance rank for females in group-living species that are not social like gorillas, like the examples reported here. As specified along the text, age and tenure are strong determinants of dominance in a species were groups are stable over multiple years. It’s just a curiosity, but do you know if there are examples of social species were female body size is associated with dominance and reproductive success?

19. Ll. 339-341. Here it is assumed that fighting abilities are correlated with body size, but aggressiveness could play an important role too maybe?

20. Ll. 377.379. Given the controversial results of other studies on the relationship between dominance rank and priority access to food resources I think some caution should be taken here.

21. Ll. 388-393. I think this is a very good idea and I agree that it should be explored once the data will be available.

22. Ll. 395-399. Same as above, only structural equation modelling (as far as I know) would allow to understand if inter-birth interval duration is indirectly influenced by energetic condition via dominance rank.

23. L. 416. Or couldn’t be the other way around too?

24. L. 427. I can see how bigger females could defend their kid from a potential predator or providing better food if they have access to better food patches, but can they really avoid infanticide? In l 214 is specified that “… neither female dominance rank or body size would likely prevent [infanticide]”.

25. Ll. 441-444. I’m not sure how body size could be correlated with longevity in gorillas. If I understood well, high-ranking females live longer than lower-ranking ones and thus have better lifetime reproductive success (Robbins et al. 2011), but no correlation between body size and dominance was found here.

26. Ll 447-449. I really appreciate the last sentence about the difficulty on the collection of data in the wild. Multiple researchers seems to fail to see that data are difficult to collect and it takes a very long time when studying long-lived species.

27. Figure 1. Personally, I do not think this figure is necessary, but I also have nothing against it. As I understand, the main message from this figure would be that female dominance in basically all groups (but ISA group) is stable over long periods of time.

28. Table 1. I have rarely seen a table cited only in the discussion, but I understand it helps developing the argument made in that section. However, since some of those results are new and belong to this study, I think they should be mentioned before in the results section.

29. Supplementary. This might be the answer to my point 2.b, but I’m not sure. “Higher-ranking females had significantly shorter inter-birth intervals than lower-ranking ones. Both back breadth and body length did not significantly influence inter-birth interval duration (Table S1). These results are similar to those found using univariate analyses presented in the text”. Does this mean that the effect of dominance rank is the in both full and univariate model?

References

Clutton-Brock TH, Huchard E. 2013. Social competition and selection in males and females. Philos. Trans. R. Soc. B Biol. Sci. 386:20130074.

Dormann CF, Elith J, Bacher S, Buchmann C, Carl G, Carré G, Marquéz JRG, Gruber B, Lafourcade B, Leitão PJ, et al. 2013. Collinearity: A review of methods to deal with it and a simulation study evaluating their performance. Ecography 36:027–046. doi:10.1111/j.1600-0587.2012.07348.x.

Grueter CC, Robbins AM, Abavandimwe D, Vecellio V, Ndagijimana F, Ortmann S, Stoinski TS, Robbins MM. 2016. Causes, mechanisms, and consequences of contest competition among female mountain gorillas in Rwanda. Behav. Ecol. 27:766–776. doi:10.1093/beheco/arv212.

Jarman PJ. 1983. Mating system and sexual dimorphism in large, terrestrial, mammalian herbivores. Biol. Rev. 58:485–520.

van de Pol M, Wright J. 2009. A simple method for distinguishing within- versus between-subject effects using mixed models. Anim. Behav. 77:753–758. doi:10.1016/j.anbehav.2008.11.006.

Rioux-Paquette E, Garant D, Martin AM, Coulson G, Festa-Bianchet M. 2015. Paternity in eastern grey kangaroos: moderate skew despite strong sexual dimorphism. Behav. Ecol. 26:1147–1155. doi:10.1093/beheco/arv052.

Robbins AM, Stoinski T, Fawcett K, Robbins MM. 2011. Lifetime reproductive success of female mountain gorillas. Am. J. Phys. Anthropol. 146:582–593. doi:10.1002/ajpa.21605.

Ronget V, Gaillard J-M, Coulson T, Garratt M, Gueyffier F, Lega J-C, Lemaître J-F. 2018. Causes and consequences of variation in offspring body mass: meta-analyses in birds and mammals. Biol. Rev. 93:1–27. doi:10.1111/brv.12329.

Shipley B. 2002. Cause and Correlation in Biology: A User’s Guide to Path Analysis, Structural Equations and Causal Inference . Bill Shipley. 1st ed. Cambridge University Press.

Wright E, Robbins AM, Robbins MM. 2014. Dominance rank differences in the energy intake and expenditure of female Bwindi mountain gorillas. Behav. Ecol. Sociobiol. 68:957–970. doi:10.1007/s00265-014-1708-9.

6. PLOS authors have the option to publish the peer review history of their article (what does this mean?). If published, this will include your full peer review and any attached files.

Reviewer #1: Yes: Emily Levy & Susan Alberts

Reviewer #2: Yes: Luca Montana

---

## [Author Response · Author response to Decision Letter 0]

31 Jan 2020

Please see Response_to_reviewers file which is the same as what is here.

PONE-D-19-28202

Dominance rank but not body size influences female reproductive success in mountain gorillas

PLOS ONE

Dear Dr. Wright,

Thank you for submitting your manuscript to PLOS ONE. After careful consideration, we feel that it has merit but does not fully meet PLOS ONE’s publication criteria as it currently stands. Therefore, we invite you to submit a revised version of the manuscript that addresses the points raised during the review process.

Both reviews appreciated the qualty of the work but also highlighted some important concerns. I would recommend to carefully consider the comments in your revision and specifically address the reviewers concerns about the statistical approach and its justification.

We would appreciate receiving your revised manuscript by Jan 16 2020 11:59PM. To enhance the reproducibility of your results, we recommend that if applicable you deposit your laboratory protocols in protocols.io, where a protocol can be assigned its own identifier (DOI) such that it can be cited independently in the future. For instructions see: http://journals.plos.org/plosone/s/submission-guidelines#loc-laboratory-protocols

• A rebuttal letter that responds to each point raised by the academic editor and reviewer(s). This letter should be uploaded as separate file and labeled 'Response to Reviewers'.

• A marked-up copy of your manuscript that highlights changes made to the original version. This file should be uploaded as separate file and labeled 'Revised Manuscript with Track Changes'.

• An unmarked version of your revised paper without tracked changes. This file should be uploaded as separate file and labeled 'Manuscript'.

We look forward to receiving your revised manuscript.

Kind regards,

Julien Martin

Academic Editor

PLOS ONE

Journal Requirements:

Reviewers' comments:

Reviewer's Responses to Questions

Comments to the Author

1. Is the manuscript technically sound, and do the data support the conclusions?

Reviewer #1: Yes

Reviewer #2: Yes

2. Has the statistical analysis been performed appropriately and rigorously? 

Reviewer #1: No

Reviewer #2: Yes

3. Have the authors made all data underlying the findings in their manuscript fully available?

Reviewer #1: Yes

Reviewer #2: Yes

4. Is the manuscript presented in an intelligible fashion and written in standard English?

Reviewer #1: Yes

Reviewer #2: Yes

5. Review Comments to the Author

Reviewer #1: This study is one of the first to identify the relationship between dominance rank, body size, and reproductive success in a wild mammal – an important question in the context of life history evolution. The authors collected two different body size measures from 34 wild female gorillas via parallel-laser photogrammetry, a method that they have used in several other publications. They hypothesize that (1) body size and dominance rank are positively associated, and (2) larger size and/or higher rank confer reproductive benefits such as shorter inter-birth intervals and higher infant survival rates.

This is an interesting and generally valid study. The authors found an association between higher rank and shorter inter-birth interval; however, they reported no association between body size and (1) dominance rank, (2) inter-birth interval, or (3) infant survival. I find the negative findings as helpful as the positive ones, and I am glad they include these three different analyses.

This is a good summary of our study and we are pleased that you recognize its importance.

Most of my suggestions are minor, and most simply serve to make the manuscript clearer for the reader. However, I have suggested a few more major changes that I think are necessary for publication and will greatly improve the introduction and data analysis. I think this study helps to fill in large gaps in our understanding of wild animal body size, especially in females, and its relationship to other demographic and life-history information.

Thanks for providing an exciting read on an important topic.

Best,

Emily Levy

Thank you for all your constructive comments which have helped to improve the manuscript. We have carefully gone through all your comments and made appropriate changes to the manuscript and added clarifications where needed. We hope that the current version satisfies all your concerns.

Major:

1. Introduction/framework

The introduction does a great job giving some relevant background information – which is a fair amount, as you must introduce literature on dominance rank, body size, reproductive success, and relationships between all of these factors. But, I think it would benefit from some re-organization and condensing. Currently there are a lot of paragraphs, each seeming to talk about something somewhat unique, which leaves the reader a bit strung out and confused and waiting to see how (and in what species) it will all come together. One suggestion is to move most of the text about the socioecological model to the discussion section titled ‘body size and dominance rank,’ when you discuss that female gorillas show low rates of aggression.

Thank you for your suggestion to improve and streamline the introduction. We have now made some considerable changes to the introduction in an effort to make it more streamlined and easier to follow.

We decided to keep the paragraph on the socioecological model in the introduction as we think it is key in explain why this topic is interesting to study in gorillas, although we did move the last sentence of this paragraph to the discussion which helps to keep it focused. We have also removed paragraph 3. 

Second, the study is framed as a test of a trade-off between adult body size and reproductive success, but such a trade-off isn’t as well-supported in the literature as the introduction makes it seem. I know of no evidence of such a trade-off in within-species studies of primates. The second paragraph discusses this trade-off, but I think most of the citations are of between-species comparative studies of primates. While these studies are helpful, patterns between species do not necessarily follow within-species (e.g., link between size & lifespan across species vs. within domestic dogs).

This idea of a trade-off is presented in the introduction and discussion. While your limited amount of energy intake must be allocated to growth, reproduction, and/or maintenance, things are a little more complicated when you consider that most growth happens before reproduction (especially in this study, where you measure females that have finished growing). So, as infants/juveniles, individuals could allocate energy to skeletal growth or maintenance but not reproduction; as adults, individuals could allocate energy to muscle mass, reproduction, or maintenance. In addition, differences in early-life environmental conditions across females in your study may also affect the extent to which they experienced any trade-offs. This is further complicated by the fact that variation in female quality may mean that some females can ‘have it all’ while others are forced to make allocation decisions. Finally, many within-species studies demonstrate that individuals who are larger also have increased fitness components (Kingsolver & Pfennig 2004 is a good meta-analysis of that). So, I think the study’s framing should be shifted away from testing for a trade-off.

We have now made considerable changes to this paragraph with your comment in mind. We still think that trade-offs is a useful framework to present the study in. We agree that we do not provide a test for this framework, but the study is not presented in that light. We merely highlight that large body size may be associated with costs and benefits. We have now added references discussing within-population trade-offs (including a primate study). We have also now stated that phenotypic variability between individuals can mask such trade-offs. 

Reference:

McLean EM, Archie EA, Alberts SC. Lifetime fitness in wild female baboons: trade-offs and individual heterogeneity in quality. Am Nat. 2019;194: 745–759.

The re-worked paragraph now reads as follows (L53):

“According to life history theory large body size is predicted to be associated with costs and benefits. Allocating energy to attain and maintain large body size is associated with delays in reproduction, as energy allocated into one fitness-enhancing function cannot be concurrently invested into another [1,2,15]. Moreover, larger females have greater absolute metabolic needs than smaller ones, which can lead to increased risk of starvation during periods of resource unpredictability [15,16]. However, whilst trade-offs are evident in some populations, variation in individual quality can mask trade-offs in others [13,17,18]. Indeed, larger individuals often have higher fitness than smaller ones [19]. The proximate mechanisms behind this may include larger females having lower infant mortality than smaller ones, owing to larger mothers producing larger offspring, more or better quality milk or providing improved offspring protection [13,15,20,21]. Moreover, large female size may be associated with advantages in resource competition, as bigger females can outcompete smaller ones and are often dominant over them [15,16]. In summary, selection has shaped individuals to be an optimal size to maximize reproductive success.”

2. Data analysis

For each gorilla, you take a mean body size and use that as the unit of analysis in the models. While this is common in the photogrammetry literature, using each photograph as the unit of analysis (and adding a random effect of individual) provides the model with within-individual variance. You could imagine a scenario in which you only have 3 photos of the two largest individuals and 10 photos of everyone else. The fewer photos you have of an individual, the smaller the probability that your mean is the true measurement of that individual. As is, your models will give equal weight to those two large individuals as everyone else. In addition, your models will not account for the fact that variance in some individual’s photos are a lot larger than in other individual’s photos. Ie, you’re more confident in the mean when SD=0.1 than when SD=1, all else being equal. I don't think you need to change the analysis for publication, but if not, you should justify your current approach.

You are right in pointing out that a mean measurement from more photos is likely to be closer to the true measurement compared with a mean from fewer photos. To ensure high accuracy of our measurements we only incorporated females with a minimum of three photos (with an average of 6 photos). Moreover, we also checked (and present) the maximum within individual CVs, to ensure that one or a few individual did not have very large within-individual variances. Lastly, we also verified that the model results are robust with regard to individual ID. We did this by re-running the models after excluding one individual at a time and comparing the estimates derived from these models to the original estimates for each predictor (model stability). The estimates were very comparable, indicating that the models are stable concerning individual ID.

We have now included several sentences in the supplementary material acknowledging this error and we present a table with the within-individual coefficients of variation (please see comment below).

There is no reporting on whether (and to what extent) the photogrammetry data enable you to differentiate between individuals. In other words, is the ratio of intra-individual error : inter-individual variance low enough for you to detect an effect of body size in the first place? One way to ask this is to ask what proportion of the variance in your body size measures are explained by individual identity. Another is to do a power analysis showing that given your sample size and the error in your photographs, you would be able to detect an effect size XX large. At the very least, a table with error/variance and/or a scatterplot with a point for each individual and SD bars around it would help the reader visualize what the data look like. Finally, it would be beneficial to indicate whether body size differed between groups, which would explain why you centered the data for one of the analyses.

Thank you for this observation, we agree that it is important to provide both within and between individual variation in body size, such that readers can assess whether we are able to differentiate females based on size. We now include a table (Table S1) detailing the within individual variation (the between-individual variation is in the original Table 1). As you will appreciate, the variance within individuals is considerably smaller than the variation between individuals, meaning that we can confidently differentiate females based on size. 

Regarding the average size of females in a given group, it did differ across groups: the lowest group average in back breadth was 47.0 and the highest was 50.4; the lowest group average in body length was 69.1 and the highest average was 73.5. We have now added a sentence in the text to this effect. Therefore statistically it makes sense to center these variables as our hypotheses concerning the effects of back breadth and body length on dominance rank were based on the relative differences between females within a group rather than absolute differences across groups. 

Minor:

On line 64, you say that “individuals should aim to find a balance among the benefits and costs of attaining optimal body size to maximize reproductive success.” I think this statement would be more accurate if written from an evolutionary perspective – it currently seems to be about individuals finding that balance, when their ability to make those decisions has (probably) been shaped by selection.

We agree that this sentence is more accurate when emphasis is placed on selection. We have rephrased it as follows:

L 66: “In summary, selection has shaped individuals to be an optimal size to maximize reproductive success.”

On line 67, you say body mass is associated with reproductive success. The next sentence states that ‘it is not always clear which component of body mass is responsible for driving such relationships.’ It’s not clear to the reader why one over another (eg, a larger skeleton vs. more muscle mass) would matter or be interesting.

We agree that this was not clear. We have now deleted this sentence and so it is no longer an issue.

On line 163 you refer the reader to your recent male body size & dominance rank manuscript for data on photogrammetry error. While that’s helpful for a general photogrammetry overview, I think you should still include the error associated with the images used in the current study (ie, within-individual %CV; see ‘major’ comment above).

We agree, and we have now included a table (Table S1) detailing the within individual error.

On line 189, you don’t specify if the time period for the mean number of dominance interactions per female. (e.g., 16 years? 1 year?)

We now include this information in the following sentence L177:

“The mean number of dominance interactions per female was 62 during an average of 10.4 years per female (range: 13 – 165; SD = 33; Fig. 1).”

Why did you switch between GLMMs and LMMs between the 1st and 2nd set of models? They seem extremely similar in construct, and glmmTMB allows several different inputs for data distributions. A brief explanation for the change would help.

We used a glmmTMB when the response was bounded between 0 and 1 (standardized dominance rank). When the response was Guassian we used a LMM. We have now highlighted that dominance rank is standardized between 0 and 1 again in this section to help readers understand that the response variables were different thus requiring different statistical approaches.

On line 258 and 266, please provide citations when discussing dispersion and stability – I’m not familiar with what an ‘allowed’ cutoff is, and readers might not be either.

We have now included citations in these sentences.

Your first model (rank ~ back breadth + body length) doesn’t include age – perhaps a citation that rank isn’t associated with age in female gorillas (or an additional model or correlation?) would justify this, as it’s common for dominance rank to change with age across both sexes of many species.

You are right, it may be important to include age in this model. We have now included age and re-run the model. The results remained unchanged.

Your description of the model results were extremely clear, and I especially appreciated that you explained the ‘effect size’ of your model coefficients. One question: On line 280, I’m confused as to why you didn’t give a separate X2 and p-value for each body size measurement individually by removing one, running log likelihood, adding it back and removing the other.

Thank you for highlighting that the results were clearly written. Before examining the significance of individual predictors one should first conduct a significance test of the full model comprising all the predictors with a null model with these predictors excluded. This is important to avoid multiple testing (Forstmeir and Schielzeth 2011). This is highlighted on line 266. As the full model was not significantly better than the null model, we concluded that all the predictors were not significant. In summary we do not gain much more useful information by examining the two predictors individually. 

On line 354, you state that back breadth %CV was ‘considerably smaller’ in females than males. However, %CV of length is larger in females than males, though the male-female difference is smaller for length than breadth. If you’re going to focus on breadth, I think you need to (1) acknowledge the length difference, and (2) run stats to show whether/that breadth %CV is appreciably larger. Also – is this the %CV reported here the average of each female’s %CV?

We have now added a sentence acknowledging that the variation in body length across females is slightly larger than it is in males, although we are unsure why this is the case. 

Concerning your comment on running stats on the %CV for back breadth to see if 4.9 is appreciably larger than 3.6. Unfortunately there is no straightforward way to statistically compare two numbers. We have deleted “considerably” from this sentence with this in mind. However, coefficients of variance are precisely that, they allow you to compare the relative amounts of variation in two variables with different means.

To provide further clarification on what these CVs are. The CVs in this table are the coefficients of variation among females (between-individual variation). This is calculated by first obtaining the mean and SD of the average measurements per female. Then we divide the SD by the mean. 

Within-individual variation, which we now present in table S1, is obtained by calculating the mean and SD for the measurements per female. Then we obtain a CV per female by dividing this SD by the mean measurement per female. Finally to obtain the average CV for within-individual variation we calculate the mean of these CVs. To answer your question, it is within-individual variation which is the average of each female’s CV but not inter-individual variation reported in this table.

In line 379-380 it sounds like there’s “little support” high rank being associated with better energy condition – does this mean there’s no support, or there’s enough to tell us about? I think fleshing this out more will be helpful, especially because you discuss the findings from Bwindi. From line 381 to the rest of the paragraph, the two ideas brought up (rank*offspring sex and future studies of IBI phases) seem like non-sequiturs after the longer discussion of rank & resource access.

We have now added the following sentence discussing the support (and lack of) the relationship between dominance rank and energetic condition in the Virunga population:

L 385: “Even though, higher-ranking females had greater access to some food resources over lower-ranking ones, dominance rank did not significantly predict energy intake or levels of urinary C-peptide, a common proxy for energy balance.”

We have also deleted the sentence discussing rank*offspring sex which we agree does not add much to the discussion. We decided to keep the sentence on IBI phases though as both us and the second reviewer thought that it is a helpful and useful mechanism to explore. 

In lines 416 and 418, it would be helpful to list the species in the studies you cite.

We have now included the species names for these references.

Lines 426 – 432, which are the start of the conclusion, would work better as a separate paragraph in the discussion. The focus of the study (in the intro) was not sexual dimorphism in gorillas.

It is true that sexual dimorphism is not the main focus of the study and we agree that these sentences should not form the start of the conclusion section. We have now made a new paragraph in the discussion section as you suggested.

Reviewer #2: This interesting manuscript seeks to understand if the unexpected relationships between dominance and different measures of female reproductive success found in previous studies from multiple mountain gorilla populations could be linked to a correlation between body size and dominance rank. Body size is normally strongly correlated to dominance rank in multiple mammalian species, granting priority access to resources that can in turn be used for increased allocation to body growth, maintenance, and to reproduction. However, this works shows that there seems to be no correlation between linear body size measures and dominance as well as with multiple indexes of reproductive success in mountain gorilla females.

This is a good summary of the study and we are happy to see that you find it interesting.

Although no effect was found, I think this study could be an interesting contribution to our knowledge on the determinants of dominance among females in sexually dimorphic species living in certain ecological and social conditions. 

We are happy to hear that you think it is a useful contribution to the literature.

I do have, however, some concerns about how the analyses were carried out and the absence of important statistical/methodological details is something that should be addressed (see below). If something is not clear or if I can be of further help please feel free to contact me (email: luca.montana@usherbrooke.ca).

Best wishes,

Luca Montana

Thank you for your helpful comments which have improved the manuscript. We have now provided more details of the statistical analyses and we hope that you are now satisfied that it is correct and that it has been done to a high standard.

Major comments

1. I understand the limitations imposed by the limited sample size on the way that statistical analyses were carried out, but I do think that it would be better if the ms compared biologically meaningful models, instead of presenting three univariate models and a full model. If multiple biologically meaningful hypotheses were translated into statisticasl models, they could be tested to understand which parameters affect female reproductive success, for example: dominance, body size, a combination of both, and possibly controlling for different parameters such as infant sex, parity etc. If this was done in model selection framework, the AIC estimated by the model selection would penalize overparametrized ones and should help understand which one is the best of the proposed models.

Whilst we agree that model selection is a valid and alternative approach, we think that the way we have analysed the data is equally good. The univariate and multivariate models reveal essentially the same results. In presenting both sets of models we can ensure that the models have enough power to reveal significant relationships (univariate models) and at the same time we can ensure that the influence of each predictor is not reliant on the presence of other predictors in the model (multivariate models). 

In addition, because we have very clear predictions in this study regarding body size, dominance rank and reproductive success, it makes sense to base our inferences on null hypothesis significance testing to test the support for these specific hypotheses. Moreover, it is unadvisable to first obtain the best model using model selection and then test the significance of a predictor due to highly inflated type 1 error (Mundry 2011). 

Reference:

Mundry 2011. Issues in information theory based statistical inference – a commentary from a frequentist’s perpective. Behavioural Ecology and Sociobiology 65: 57 – 68.

2. Related to the my comment above, this specifically applies to both inter-birth interval and infant survival predictions:

a. Why are back breadth and body length not joined in the same ‘body size’ model? Since both measures describe linear body size, I’m not sure I understand why they are not included in one multivariate model testing if body size has an effect on dominance, inter-birth interval duration, or infant survival. Using the rule of thumb described in Dormann et al. (2013) two variables correlated by rs < 0.7 can be used in the same model.

In both the inter-birth interval and infant survival models we did fit multivariate models which included back breadth and body length in addition to other variables (supplementary material). However, due to the low number of data points in these models, they likely had very low power. Due to this we also fitted univariate models to test the variables of interest. 

To clarify it was not because back breadth and body length were somewhat (albeit weakly) correlated. You are right, correlated variables by rs < 0.7 can usually be included in the same model. But due to the low number of data points and resulting low power.

To ensure that the conclusions resulting from the univariate models are valid we also fitted multivariate models (supplementary material) with all the relevant predictors, to verify that the influence of each predictor is not dependent on other predictors being present in the model. As both sets of models revealed essentially the same results we can be confident in the results from the univariate models presented in the main text.

Also please see answer to comment 29.

b. I think it would be interesting to compare the effects of dominance rank on inter-birth interval duration of the univariate model vs the full model. Although the full model is not significantly different from the null model, dominance is significant in both models. If the effect of dominance is different between the two models, this could mean that other parameters influence the effect of dominance on reproductive success and thus the univariate model it’s not the best way to make predictions.

Thanks for pointing this out. Please note that the effect of dominance rank on inter-birth intervals is very similar in the univariate and multivariate models as can be seen from the estimates for dominance rank in the two models (estimate ± se: -2.57 ± 1.21 and -3.18 ± 1.36, respectively). These results indicate that the influence of dominance rank on inter-birth interval does not depend on other predictors being present in the model and the results from the univariate model are valid.

3. I think the ms would benefit from more information about the choice of the models used to test the interesting hypotheses presented here. I specifically pointed out some examples in my minor comments

We hope we have now convinced you that our approach to the analyses is valid, and indeed may be superior to model selection for example. In addition, we have now included additional explanations in the text to improve clarity on why we fitted both univariate and multivariate models testing the same hypothesis.

Minor comments

4. L. 67-68. This is really petty and I apologize to pointing it out, but the sentence says that body mass is positively correlated with female reproductive success in mammal, but then a study on both birds and mammals is cited. I did not go through the cited paper (Ronget et al. 2018), but is the correlation showed just on mammal or on both birds and mammals?

The cited study is a meta-analyses and finds positive correlations between body mass and reproductive success in both mammals and birds, although the correlation is stronger in mammals. 

Although, we have now deleted this paragraph and so this is no longer an issue. 

5. L. 73. I think something is either missing or the article ‘a’ should be removed from the sentence.

Thank you for highlighting this typo, we have now delete this sentence with the restructuring of the introduction.

6. Ll 107-109. In its current state, the sentence argues that body size is correlated with female dominance only in social species. Wouldn’t that be the case also for group-living but not strictly-social species?

Thank you for highlighting this. We have now changed social to group-living species here.

7. Ll 120-122. I admit I was surprised by the fact that gorillas had one of the highest degree of male biased ssd in mammal. I thought other mammalian species showed higher degrees than that (not true: Jarman (1983)), but I found out that many times it’s not the average that is reported (e.g. Rioux-Paquette et al. (2015): males can weight up to 4 times as much as the smallest breeding female). Today I learned multiple new things!

We are happy to hear that you found this information intriguing.

8. Ll. 126-129. In my opinion, this is a critical part of the introduction but I find it sometimes confusing. May I suggest to spend some more words to better introduce the correlation between dominance and priority of access to some food resources etc? Also, I might be wrong, but I think that it was Wright et al. (2014) that described the decreased travelling time? So reference 64 instead of 61.

Higher-ranking females spent longer times feeding on foods which are easier to contest over (or usurp) than lower-ranking females (Grueter et al. 2016). But we think that the sentence in the text is easier to understand: “Higher-ranking females have priority of access to some food resources…”. 

Moreover, we introduce the socioecological theory behind the general relationship between dominance rank and access to resources in the previous paragraph. We have also included a sentence in the discussion providing more detail:

L 385: “Even though, higher-ranking females had greater access to some food resources over lower-ranking ones, dominance rank did not significantly predict energy intake or levels of urinary C-peptide, a common proxy for energy balance.”

You are correct, reference 64 should replace reference 61, which we have now corrected.

9. Ll. 164-168. At first I started looking for the sample size here, and I realized only later that N changed depending on the analyses. Maybe it could be a good idea to specify it here? Also, related to N, since repeated measurements of the same individuals are present, I think it could be a good idea to specify both the total number of measurement and the number of females measured.

You are right, not all females were used in all analyses. For example, in the inter-birth interval model we could only include females that gave birth to at least two offspring of which the first offspring survived to weaning age, which resulted in some females being excluded. In the original manuscript we stated the number of females in the study on L154. Moreover, we have also included the number of females that each analysis incorporates (in addition to the number of data points).

10. Ll. 171-189. I really appreciate the details on how dominance was estimated. Good job! However, related to the next comment and to comment n. 3, only by looking at Figure 2 I understood that ranking was treated as a continuous variables that ranges from 0 to 1. I think it would be appropriate to specify how this was done, because by looking at Figure 1 Elo-rating values range between -300 and about 1350. Also, was dominance rank standardized or scaled between 0 and 1? It’s not clear to me how the values I just mentioned could be standardized between 0 and 1.

We are happy to hear that you felt that the methods pertaining to estimating dominance were clear and well presented. In the original text we included a sentence explaining how rank was standardized on line 186. Dominance rank was standardized not scaled. Meaning that the highest rank in a group on a given day was given a 1 and the lowest a 0, with the ranks in between were set proportional to the Elo rating. To do this we employed the standardize argument in the extract.elo function from the EloRating package. We have now highlighted this again in the original sentence that dominance rank is standardized between 0 and 1; see below.

11. L. 218. I did not understand how a beta distribution could be used to test if back breadth and body length were correlated with dominance rank until I saw that dominance ranks were a continuous variable ranging between 0 and 1.

We have now added to the original sentence highlighting that dominance rank was standardized between 0 and 1 again here.

12. Ll. 222-223. I think a small sentence explaining why the random slopes of back breadth and body length were used within group ID (which I think means ‘back breadth | group ID’ and ‘body length | group ID’). I can see why the effect of body measurements could change depending on the group, but I think readers could benefit from a short sentence on why these random slopes were used.

The inclusion of random slopes is standard when possible (Barr et al. 2013). We included a citation in the original sentence in case readers are not familiar with random slopes or why they are included.

13. L. 227. Not sure about the following reference: van de Pol and Wright (2009). I think it could mislead people looking for papers using subject centering. I think the explanation in ll. 223-227 is really clear and I agree that this was the best way to see if morphological measures had an effect on dominance rank based on relative differences between females within the same group.

We are glad you agree on the statistical approach here. To increase clarity we have excluded the van de Pol and Wright reference. Also, we now highlight the differences in group means to further highlight this.

14. Ll. 239-240. Since I’m not working with a species that forms stable groups it took me a while to understand the “…the mother transferred in between the birth of two infants…” meant that females could move to different groups between giving births. I think it could benefit the readers to add a sentence when talking about the natural history on the species.

Gorillas live in cohesive stable social units, but females exhibit secondary dispersal, meaning that adult females may transfer between groups when they do not have unweaned offspring.

By providing four references in this sentence, we hope readers can follow up on these points if necessary.

15. Ll. 247-250. I’m no expert of Cox proportional hazardous models, but aren’t they used to make prediction when using censored data? I do not understand if censored data are present (related to comment n. 3), and if not why Cox regression were the model of choice. Instead of a Cox regression, wouldn’t a binomial glmm have been enough to estimate the probability that a young reached weaning age?

Both a binomial glmm and a Cox regression are possible approaches here, but the Cox model is more informative. The data in the mentioned model do include censored data (number of days until weaning or death). You are right, we could have also used a binomial response with 0/1 for infant survival to weaning. However, by using a Cox model we gain information lost in the binomial model, namely the number of days when the infant dies (when they do not reach weaning age). Theoretically this could be useful information. We checked to see if both models revealed different results and, as expected, both sets of results are very similar. 

16. Ll. 270-271. Given what is stated in ll 223-227 (centering of back breadth and body length among females within each group to a mean of zero), I assume that these two measurements were z-transformed too, am I right? This would mean that back breadth and body length were z-transformed differently depending on the hypothesis that was being tested. If this is the case, I think it could help readers if something was said about this matter either here or at ll 223-227.

You are correct, the centered back breadth and body length variables were then z-transformed. We have now added that this z-transformation was done per analysis on L266. 

17. Lines 320-323. Given the controversial results on different populations of the same species I think here there are lots of assumptions that are used to say that rank may influence body condition (or the other way around?), which may be in turn a stronger correlate of reproductive success than linear body size. The only way to test this would be through structural equation modelling (Shipley 2002). Similar comments below.

This study adds to the discussion and we are only suggesting that rank may influence female body condition (even if there are other factors that cause rank to influence body condition such as better access to resources).

18. Ll. 329-332. There are lots of contradictory studies on the correlation dominance rank-body size in females (Clutton-Brock and Huchard 2013). It appears to me that body size was shown to correlate with dominance rank for females in group-living species that are not social like gorillas, like the examples reported here. As specified along the text, age and tenure are strong determinants of dominance in a species were groups are stable over multiple years. It’s just a curiosity, but do you know if there are examples of social species were female body size is associated with dominance and reproductive success?

Unfortunately very few studies have data on body size, therefore this is generally poorly known. There is some indication in rhesus macaques (Blomquist and Turnquist 2011) and in slightly less social species such as fur seals (Beauplet and Guinet 2007) and elephant seals (Reiter, Panken, Le Boeuf 1981).

References:

Blomquist GE, Turnquist JE. Selection on adult female body size in rhesus macaques. J Hum Evol. 2011;60: 677–683. doi:10.1016/j.jhevol.2010.05.010

Beauplet G, Guinet C. Phenotypic determinants of individual fitness in female fur seals: larger is better. Proc R Soc Lond B Biol Sci. 2007;274: 1877–1883. doi:10.1098/rspb.2007.0454

Reiter J, Panken KJ, Le Boeuf BJ. Female competition and reproductive success in northern elephant seals. Anim Behav. 1981;29: 670–687. doi:10.1016/S0003-3472(81)80002-4

19. Ll. 339-341. Here it is assumed that fighting abilities are correlated with body size, but aggressiveness could play an important role too maybe?

We agree that aggressiveness could potentially be important as well. We have added “levels of aggression” to this sentence.

20. Ll. 377.379. Given the controversial results of other studies on the relationship between dominance rank and priority access to food resources I think some caution should be taken here.

Thanks for the suggestion of being cautious. Even though we agree that there is controversy regarding the relationship between dominance rank and priority of access to resources, it remains the most widely used explanation for positive relationships between dominance rank and reproductive success. Therefore, we think that this is a valid statement. We are simply stating that our results are similar to those of some other studies. We are not stating that this is a standard pattern across mammals, social species, etc.

21. Ll. 388-393. I think this is a very good idea and I agree that it should be explored once the data will be available.

We are happy to hear that you agree.

22. Ll. 395-399. Same as above, only structural equation modelling (as far as I know) would allow to understand if inter-birth interval duration is indirectly influenced by energetic condition via dominance rank.

Unfortunately, this is beyond the scope of the current study. But we thank you for the tip. Again, we are simply suggesting that this may be the causal pattern, but as in most correlational studies, we can only state what we think may be the driving mechanism.

23. L. 416. Or couldn’t be the other way around too?

Yes, we agree that better body condition can lead to higher dominance rank rather than higher dominance rank leading to better body condition. We included a sentence to this effect on L341 of the original manuscript. Again, we are suggesting that body size influences rank, but since the study is correlational, we can’t do more than that. 

24. L. 427. I can see how bigger females could defend their kid from a potential predator or providing better food if they have access to better food patches, but can they really avoid infanticide? In l 214 is specified that “… neither female dominance rank or body size would likely prevent [infanticide]”.

We agree that it is unlikely that in gorillas larger females are able to better defend their infants from infanticide than smaller females. However, more generally as in baboons for example, the risk of infanticide was found to be dependent on female rank, therefore potentially on body size as well (Cheney et al. 2004). 

Reference:

Cheney DL, Seyfarth RM, Fischer J, Beehner J, Bergman T, Johnson SE, et al. Factors affecting reproduction and mortality among baboons in the Okavango Delta, Botswana. Int J Primatol. 2004;25: 401–428. doi:10.1023/B:IJOP.0000019159.75573.13

25. Ll. 441-444. I’m not sure how body size could be correlated with longevity in gorillas. If I understood well, high-ranking females live longer than lower-ranking ones and thus have better lifetime reproductive success (Robbins et al. 2011), but no correlation between body size and dominance was found here.

We think that just because body size does not correlate with dominance rank, and dominance rank does correlate with longevity this does not have to mean that body size is not correlated with longevity, especially if there are some other benefits of larger body size that haven’t been taken into account. 

26. Ll 447-449. I really appreciate the last sentence about the difficulty on the collection of data in the wild. Multiple researchers seems to fail to see that data are difficult to collect and it takes a very long time when studying long-lived species.

We could not agree more and are happy to see you feel the same.

27. Figure 1. Personally, I do not think this figure is necessary, but I also have nothing against it. As I understand, the main message from this figure would be that female dominance in basically all groups (but ISA group) is stable over long periods of time.

Yes, that is correct. We would like to retain it so show the stability over time.

28. Table 1. I have rarely seen a table cited only in the discussion, but I understand it helps developing the argument made in that section. However, since some of those results are new and belong to this study, I think they should be mentioned before in the results section.

We agree and we now present the table in the results section.

29. Supplementary. This might be the answer to my point 2.b, but I’m not sure. “Higher-ranking females had significantly shorter inter-birth intervals than lower-ranking ones. Both back breadth and body length did not significantly influence inter-birth interval duration (Table S1). These results are similar to those found using univariate analyses presented in the text”. Does this mean that the effect of dominance rank is the in both full and univariate model?

Yes you are correct. In essence the multivariate model results are essentially the same as the univariate model results. But due to the increased model complexity the models have low power. We think that presenting both the univariate and multivariate models is the best compromise concerning the inclusion of all potentially relevant predictors and low sample size and related low power. We agree with your point above, model selection would have been an equally valid alternative approach. The downside of model selection is that it is then unadvisable to then test the significance of individual predictors (obtain a p-values) due to multiple testing (Mundry 2011). We have confidence that model selection would reveal the same results reported here, but without p-values.

References

Clutton-Brock TH, Huchard E. 2013. Social competition and selection in males and females. Philos. Trans. R. Soc. B Biol. Sci. 386:20130074.

Dormann CF, Elith J, Bacher S, Buchmann C, Carl G, Carré G, Marquéz JRG, Gruber B, Lafourcade B, Leitão PJ, et al. 2013. Collinearity: A review of methods to deal with it and a simulation study evaluating their performance. Ecography 36:027–046. doi:10.1111/j.1600-0587.2012.07348.x.

Grueter CC, Robbins AM, Abavandimwe D, Vecellio V, Ndagijimana F, Ortmann S, Stoinski TS, Robbins MM. 2016. Causes, mechanisms, and consequences of contest competition among female mountain gorillas in Rwanda. Behav. Ecol. 27:766–776. doi:10.1093/beheco/arv212.

Jarman PJ. 1983. Mating system and sexual dimorphism in large, terrestrial, mammalian herbivores. Biol. Rev. 58:485–520.

van de Pol M, Wright J. 2009. A simple method for distinguishing within- versus between-subject effects using mixed models. Anim. Behav. 77:753–758. doi:10.1016/j.anbehav.2008.11.006.

Rioux-Paquette E, Garant D, Martin AM, Coulson G, Festa-Bianchet M. 2015. Paternity in eastern grey kangaroos: moderate skew despite strong sexual dimorphism. Behav. Ecol. 26:1147–1155. doi:10.1093/beheco/arv052.

Robbins AM, Stoinski T, Fawcett K, Robbins MM. 2011. Lifetime reproductive success of female mountain gorillas. Am. J. Phys. Anthropol. 146:582–593. doi:10.1002/ajpa.21605.

Ronget V, Gaillard J-M, Coulson T, Garratt M, Gueyffier F, Lega J-C, Lemaître J-F. 2018. Causes and consequences of variation in offspring body mass: meta-analyses in birds and mammals. Biol. Rev. 93:1–27. doi:10.1111/brv.12329.

Shipley B. 2002. Cause and Correlation in Biology: A User’s Guide to Path Analysis, Structural Equations and Causal Inference . Bill Shipley. 1st ed. Cambridge University Press.

Wright E, Robbins AM, Robbins MM. 2014. Dominance rank differences in the energy intake and expenditure of female Bwindi mountain gorillas. Behav. Ecol. Sociobiol. 68:957–970. doi:10.1007/s00265-014-1708-9.

6. PLOS authors have the option to publish the peer review history of their article (what does this mean?). If published, this will include your full peer review and any attached files.

Do you want your identity to be public for this peer review? For information about this choice, including consent withdrawal, please see our Privacy Policy.

Reviewer #1: Yes: Emily Levy & Susan Alberts

Reviewer #2: Yes: Luca Montana

---

## [Decision Letter · Decision Letter 1]

27 Feb 2020

PONE-D-19-28202R1

Dominance rank but not body size influences female reproductive success in mountain gorillas

PLOS ONE

Dear Dr. Wright,

Thank you for submitting your manuscript to PLOS ONE. After careful consideration, we feel that it has merit but does not fully meet PLOS ONE’s publication criteria as it currently stands. Therefore, we invite you to submit a revised version of the manuscript that addresses the points raised during the review process.

Both reviewers and myself found that you dealt adequately with the previous comments and resubmit a stronger version of the manuscript. However, both reviewers have some minor suggestions to further improve the clarity of the manuscript. 

We would appreciate receiving your revised manuscript by Apr 12 2020 11:59PM. To enhance the reproducibility of your results, we recommend that if applicable you deposit your laboratory protocols in protocols.io, where a protocol can be assigned its own identifier (DOI) such that it can be cited independently in the future. For instructions see: http://journals.plos.org/plosone/s/submission-guidelines#loc-laboratory-protocols

We look forward to receiving your revised manuscript.

Kind regards,

Julien Martin

Academic Editor

PLOS ONE

Reviewers' comments:

Reviewer's Responses to Questions

**Comments to the Author**

1. If the authors have adequately addressed your comments raised in a previous round of review and you feel that this manuscript is now acceptable for publication, you may indicate that here to bypass the “Comments to the Author” section, enter your conflict of interest statement in the “Confidential to Editor” section, and submit your "Accept" recommendation.

Reviewer #1: (No Response)

Reviewer #2: (No Response)

2. Is the manuscript technically sound, and do the data support the conclusions?

Reviewer #1: Yes

Reviewer #2: Yes

3. Has the statistical analysis been performed appropriately and rigorously? 

Reviewer #1: Yes

Reviewer #2: Yes

4. Have the authors made all data underlying the findings in their manuscript fully available?

Reviewer #1: Yes

Reviewer #2: Yes

5. Is the manuscript presented in an intelligible fashion and written in standard English?

Reviewer #1: Yes

Reviewer #2: Yes

6. Review Comments to the Author

Reviewer #1: Thank you for your thoughtful editing of this manuscript. I have a few smaller comments remaining. I think all major issues have been addressed.

Abstract:

The sentence on lines 22-26 is very long and the ideas aren’t connected in a way that’s easy to understand. Creating two sentences and clarifying would do the trick. I think you’re saying that (1) Previous studies saw a relationship between rank and reproduction, (2) that’s surprising because their hierarchy is relatively weak, (3) maybe hierarchy is weak because feeding competition is weak, (4) maybe rank is associated with body size because...[of the association between rank and reproduction?]

Line 31: You say ‘no support for body size influencing…’ but I don’t think you can assume causation. The word ‘influence’ is used in line 34 as well.

Introduction:

The first four paragraphs don’t connect or ‘flow’ together well, so it is challenging to read. Perhaps the first sentence of each paragraphs 2-4 should include a transition from the previous paragraph’s topic to the current paragraph’s topic to help the reader.

Line 66-67: Thanks for shifting this sentence around. One more suggestion is that ‘optimal’ isn’t necessarily right. Not only is selection acting on the whole organism (not just body size), but selection isn’t an optimizing process.

Line 121-122: If you shift the parenthetical to the end of the sentence it will be easier to read.

Line 124: It’s not yet clear why the authors thought the rank-reproduction association might be attributable to a rank-body size association – spelling it out clearly will help lead the reader to your present study

Line 134: You already know that rank is associated with these reproductive measures in this population, right? If so, perhaps include that as a citation and explain why you’re testing it again.

Results:

Line 283: It’s helpful for the reader to briefly give the formula for %CV for those unfamiliar with it.

Thanks for adding in the CVs for each individual (Table S1). I didn’t see that table directly referenced in the text; perhaps it could be referenced in the caption for Table 1 (and sorry if I missed it elsewhere).

Line 317: Given your confidence interval for back breadth, I don’t think it’s helpful to mention this last sentence.

Discussion:

Line 385: There’s an extra comma after “Even though”

Line 418: Specify whether the rank-reproduction relationship in chimpanzees is in females specifically.

Reviewer #2: I would like to praise the authors for their constructive responses and for the changes they have made following the comments from both revisions. Although I am normally more in favor of an information-theoretic approach, data do not always justify or allow its use. Since it was repeated multiple times in the authors’ response to reviewers that the sample size had low power, I run a couple of power analyses myself to test what was the power given by the available dataset and different R2 values. I thus agree that the statistical approach used here is valid and justified by the nature and amount of available data. I appreciate the improved clarity on the choice of univariate vs multivariate models.

I highlight here a few possible minor changes, but I otherwise think this article is a sound and fascinating work that will be received with interested by the ecological community.

Best wishes,

Luca Montana

Minor comments

1. Ll 66-67. I would suggest to use the present tense, so “[…] selection shapes individuals to be an optimal size to maximize reproductive success”. In its present form, it seems like individuals already attained the perfect size to maximise their fitness. If that was the case, the variance on a trait such as body size would be greatly reduced.

2. Ll. 230-232. I think it would be helpful to disclose the link function of the beta model used to test the first hypothesis. I assume it was a logit function?

3. Ll 275-276. Should this paragraph be included in a specific section like Ethical statement or something like it?

4. Ll 329-330. Is there any reference that would support this assumption? Probably Grueter et al. (2016), Wright and Robbins (2014) or one among the references cited in l 379? This is merely because if somebody jumps directly to the discussion the absence of references could suggest a lack of knowledge to sustain this, although logical, assumption.

5. Ll 384-388. I wonder if the comma after ‘even though’ in the following sentence is appropriate “Even though, higher-ranking females had greater access to some food resources over lower-ranking ones, dominance rank did not significantly predict energy intake rate or levels of urinary C-peptide, a common proxy for energy balance”. I am not totally sure since I’m not a native english speaker,

6. Ll. 435-441. I wonder if this last paragraph is relevant to the discussion. Sexual size dimorphism is only briefly cited in the introduction, and given that selection on body size was not formally tested here I would suggest to remove it entirely.

7. Supplementary material. I think that the appropriate reference for this sentence between parentheses “keeping in mind the risk of multiple testing” was actually 2 (Forstmeier and Schielzeth 2011) and not 1 (Wright et al. 2019).

References

Forstmeier W, Schielzeth H. 2011. Cryptic multiple hypotheses testing in linear models: Overestimated effect sizes and the winner’s curse. Behav Ecol Sociobiol 65:47–55. doi:10.1007/s00265-010-1038-5.

Grueter CC, Robbins AM, Abavandimwe D, Vecellio V, Ndagijimana F, Ortmann S, Stoinski TS, Robbins MM. 2016. Causes, mechanisms, and consequences of contest competition among female mountain gorillas in Rwanda. Behav Ecol 27:766–776. doi:10.1093/beheco/arv212.

Wright E, Galbany J, McFarlin SC, Ndayishimiye E, Stoinski TS, Robbins MM. 2019. Male body size, dominance rank and strategic use of aggression in a group-living mammal. Anim Behav 151:87–102. doi:10.1016/j.anbehav.2019.03.011.

Wright E, Robbins MM. 2014. Proximate mechanisms of contest competition among female Bwindi mountain gorillas (Gorilla beringei beringei). Behav Ecol Sociobiol 68:1785–1797. doi:10.1007/s00265-014-1788-6.

7. PLOS authors have the option to publish the peer review history of their article (what does this mean?). If published, this will include your full peer review and any attached files.

Reviewer #1: Yes: Emily Levy

Reviewer #2: Yes: Luca Montana

---

## [Author Response · Author response to Decision Letter 1]

8 Apr 2020

PONE-D-19-28202R1

Dominance rank but not body size influences female reproductive success in mountain gorillas

PLOS ONE

Dear Dr. Wright,

Thank you for submitting your manuscript to PLOS ONE. After careful consideration, we feel that it has merit but does not fully meet PLOS ONE’s publication criteria as it currently stands. Therefore, we invite you to submit a revised version of the manuscript that addresses the points raised during the review process.

Both reviewers and myself found that you dealt adequately with the previous comments and resubmit a stronger version of the manuscript. However, both reviewers have some minor suggestions to further improve the clarity of the manuscript. 

We would appreciate receiving your revised manuscript by Apr 12 2020 11:59PM. To enhance the reproducibility of your results, we recommend that if applicable you deposit your laboratory protocols in protocols.io, where a protocol can be assigned its own identifier (DOI) such that it can be cited independently in the future. For instructions see: http://journals.plos.org/plosone/s/submission-guidelines#loc-laboratory-protocols

• A rebuttal letter that responds to each point raised by the academic editor and reviewer(s). This letter should be uploaded as separate file and labeled 'Response to Reviewers'.

• A marked-up copy of your manuscript that highlights changes made to the original version. This file should be uploaded as separate file and labeled 'Revised Manuscript with Track Changes'.

• An unmarked version of your revised paper without tracked changes. This file should be uploaded as separate file and labeled 'Manuscript'.

We look forward to receiving your revised manuscript.

Kind regards,

Julien Martin

Academic Editor

PLOS ONE

Dear Julien,

We are happy to hear that both reviewers and you were happy with the changes we made and with the revised manuscript. We have now carefully gone through these additional minor comments and hope that you will now be satisfied with the current version. 

Many thanks.

Kind regards,

Edward Wright

Reviewers' comments:

Reviewer's Responses to Questions

Comments to the Author

1. If the authors have adequately addressed your comments raised in a previous round of review and you feel that this manuscript is now acceptable for publication, you may indicate that here to bypass the “Comments to the Author” section, enter your conflict of interest statement in the “Confidential to Editor” section, and submit your "Accept" recommendation.

Reviewer #1: (No Response)

Reviewer #2: (No Response)

2. Is the manuscript technically sound, and do the data support the conclusions?

Reviewer #1: Yes

Reviewer #2: Yes

3. Has the statistical analysis been performed appropriately and rigorously? 

Reviewer #1: Yes

Reviewer #2: Yes

4. Have the authors made all data underlying the findings in their manuscript fully available?

Reviewer #1: Yes

Reviewer #2: Yes

5. Is the manuscript presented in an intelligible fashion and written in standard English?

Reviewer #1: Yes

Reviewer #2: Yes

6. Review Comments to the Author

Reviewer #1: Thank you for your thoughtful editing of this manuscript. I have a few smaller comments remaining. I think all major issues have been addressed.

We were pleased to hear that you were much happier with the revised manuscript. We have now carefully gone through these additional comments and we hope that you will now be fully satisfied with the current version. We thank you once again for your efforts in helping us improve this manuscript.

Abstract:

The sentence on lines 22-26 is very long and the ideas aren’t connected in a way that’s easy to understand. Creating two sentences and clarifying would do the trick. I think you’re saying that (1) Previous studies saw a relationship between rank and reproduction, (2) that’s surprising because their hierarchy is relatively weak, (3) maybe hierarchy is weak because feeding competition is weak, (4) maybe rank is associated with body size because...[of the association between rank and reproduction?]

We agree that this sentence could be improved and we have now rephrased it as follows:

Previous studies found that female dominance rank correlates with reproductive success in mountain gorillas (Gorilla beringei beringei), which is surprising given they have weak dominance relationships and experience seemingly low levels of feeding competition. It is not currently known whether this relationship is primarily driven by a positive correlation between rank and body size.

Line 31: You say ‘no support for body size influencing…’ but I don’t think you can assume causation. The word ‘influence’ is used in line 34 as well.

We agree that we cannot assume causation here and we have now replaced “influence” with “correlation”. The two new sentences are as follows:

“Using linear mixed models, we found no support for body size to be significantly correlated with dominance rank or female reproductive success. Higher-ranking females had significantly shorter inter-birth intervals than lower-ranking ones, but dominance rank was not significantly correlated with infant mortality.”

Introduction:

The first four paragraphs don’t connect or ‘flow’ together well, so it is challenging to read. Perhaps the first sentence of each paragraphs 2-4 should include a transition from the previous paragraph’s topic to the current paragraph’s topic to help the reader.

We have rewritten the first sentences of paragraphs 3, 4, and 5 to improve the transitions among paragraphs and improve the readability of these paragraphs. We realize we are covering a lot of ground in these few paragraphs and the linkages may not be so obvious to all readers. Thank you for pointing this out.

Line 66-67: Thanks for shifting this sentence around. One more suggestion is that ‘optimal’ isn’t necessarily right. Not only is selection acting on the whole organism (not just body size), but selection isn’t an optimizing process.

We have now replaced the word “optimal’ with “ideal”.

Line 121-122: If you shift the parenthetical to the end of the sentence it will be easier to read.

We have now placed the parenthesis at the end of the sentence.

Line 124: It’s not yet clear why the authors thought the rank-reproduction association might be attributable to a rank-body size association – spelling it out clearly will help lead the reader to your present study

We have now rephrased this sentence and we think it is now clearer to understand:

Given the low levels of feeding competition and weak dominance relationships, such relationships were not expected and those authors suggested that the positive correlation between dominance rank and reproductive success may in fact be a by-product of a positive correlation between rank and body size, such that body size is driving the relationship, not rank. 

Line 134: You already know that rank is associated with these reproductive measures in this population, right? If so, perhaps include that as a citation and explain why you’re testing it again.

We are testing it again because we have a different data set and we believe it is useful to also test if the relationship between rank and reproductive success still holds in this population, especially since it was not expected.

Results:

Line 283: It’s helpful for the reader to briefly give the formula for %CV for those unfamiliar with it.

We have now included the formula for %CV in this table.

Thanks for adding in the CVs for each individual (Table S1). I didn’t see that table directly referenced in the text; perhaps it could be referenced in the caption for Table 1 (and sorry if I missed it elsewhere).

Thank you for highlighting this. We have now referenced Table S1 in the caption of Table 1 to refer readers to intra-individual CVs.

Line 317: Given your confidence interval for back breadth, I don’t think it’s helpful to mention this last sentence.

We agree, and we have now deleted this sentence. 

Discussion:

Line 385: There’s an extra comma after “Even though”

We have deleted this comma, thank you.

Line 418: Specify whether the rank-reproduction relationship in chimpanzees is in females specifically.

We now make it clear that this relationship is present in females specifically.

Reviewer #2: I would like to praise the authors for their constructive responses and for the changes they have made following the comments from both revisions. Although I am normally more in favor of an information-theoretic approach, data do not always justify or allow its use. Since it was repeated multiple times in the authors’ response to reviewers that the sample size had low power, I run a couple of power analyses myself to test what was the power given by the available dataset and different R2 values. I thus agree that the statistical approach used here is valid and justified by the nature and amount of available data. I appreciate the improved clarity on the choice of univariate vs multivariate models.

I highlight here a few possible minor changes, but I otherwise think this article is a sound and fascinating work that will be received with interested by the ecological community.

Best wishes,

Luca Montana

We are happy to hear that you were satisfied with our changes and that you agree with our statistical approach. Thank you for your additional suggestions. We have now carefully gone through these and made the appropriate changes. Thank you once again for your efforts in helping to improve this manuscript. 

Minor comments

1. Ll 66-67. I would suggest to use the present tense, so “[…] selection shapes individuals to be an optimal size to maximize reproductive success”. In its present form, it seems like individuals already attained the perfect size to maximise their fitness. If that was the case, the variance on a trait such as body size would be greatly reduced.

Thank you for this suggestion. We have now changed this sentence to the present tense and replaced the word ‘optimal’.

2. Ll. 230-232. I think it would be helpful to disclose the link function of the beta model used to test the first hypothesis. I assume it was a logit function?

Yes, you are correct it is a logit function which we now specify in the text.

3. Ll 275-276. Should this paragraph be included in a specific section like Ethical statement or something like it?

We have now added “Ethical Note” above this paragraph.

4. Ll 329-330. Is there any reference that would support this assumption? Probably Grueter et al. (2016), Wright and Robbins (2014) or one among the references cited in l 379? This is merely because if somebody jumps directly to the discussion the absence of references could suggest a lack of knowledge to sustain this, although logical, assumption.

Thank you for this suggestion. We have now added both Grueter et al. 2016 and Wright et al. 2014 citations to this sentence.

5. Ll 384-388. I wonder if the comma after ‘even though’ in the following sentence is appropriate “Even though, higher-ranking females had greater access to some food resources over lower-ranking ones, dominance rank did not significantly predict energy intake rate or levels of urinary C-peptide, a common proxy for energy balance”. I am not totally sure since I’m not a native english speaker,

We have now deleted this comma.

6. Ll. 435-441. I wonder if this last paragraph is relevant to the discussion. Sexual size dimorphism is only briefly cited in the introduction, and given that selection on body size was not formally tested here I would suggest to remove it entirely.

Thank you for this suggestion to streamline this section of the discussion. Whilst we agree that sexual size dimorphism does not form a central part of the study, we still think it is a useful short section to include for two reasons. 1) we do provide and discuss the equivalent data on male gorillas for comparative value and 2) this section helps to better understand why females are considerably smaller than males overall. As this is a small section we do not think it distracts the readers from the main message of the study. 

7. Supplementary material. I think that the appropriate reference for this sentence between parentheses “keeping in mind the risk of multiple testing” was actually 2 (Forstmeier and Schielzeth 2011) and not 1 (Wright et al. 2019).

You are right, thank you very much for highlighting this mistake.

References

Forstmeier W, Schielzeth H. 2011. Cryptic multiple hypotheses testing in linear models: Overestimated effect sizes and the winner’s curse. Behav Ecol Sociobiol 65:47–55. doi:10.1007/s00265-010-1038-5.

Grueter CC, Robbins AM, Abavandimwe D, Vecellio V, Ndagijimana F, Ortmann S, Stoinski TS, Robbins MM. 2016. Causes, mechanisms, and consequences of contest competition among female mountain gorillas in Rwanda. Behav Ecol 27:766–776. doi:10.1093/beheco/arv212.

Wright E, Galbany J, McFarlin SC, Ndayishimiye E, Stoinski TS, Robbins MM. 2019. Male body size, dominance rank and strategic use of aggression in a group-living mammal. Anim Behav 151:87–102. doi:10.1016/j.anbehav.2019.03.011.

Wright E, Robbins MM. 2014. Proximate mechanisms of contest competition among female Bwindi mountain gorillas (Gorilla beringei beringei). Behav Ecol Sociobiol 68:1785–1797. doi:10.1007/s00265-014-1788-6.

7. PLOS authors have the option to publish the peer review history of their article (what does this mean?). If published, this will include your full peer review and any attached files.

Do you want your identity to be public for this peer review? For information about this choice, including consent withdrawal, please see our Privacy Policy.

Reviewer #1: Yes: Emily Levy

Reviewer #2: Yes: Luca Montana

---

## [Editor Report · Decision Letter 2]

23 Apr 2020

PONE-D-19-28202R2

Dominance rank but not body size influences female reproductive success in mountain gorillas

PLOS ONE

Dear Dr. Wright,

Thank you for submitting your manuscript to PLOS ONE.  I am really pleased with the manuscript and I would ask for just 1 minor modification before final acceptance. In addition I would suggest that you take the time to carefully read the manuscript, remove any hidden text and hidden comments from the word file. Unfortunately, the editing process in PLOS ONE would integrate the hidden text and comments in the final published pdf and there is no proof editing with PLOS ONE. So this is your last opportunity to make minor text edits to the manuscript before publications.

We would appreciate receiving your revised manuscript by Jun 07 2020 11:59PM. To enhance the reproducibility of your results, we recommend that if applicable you deposit your laboratory protocols in protocols.io, where a protocol can be assigned its own identifier (DOI) such that it can be cited independently in the future. For instructions see: http://journals.plos.org/plosone/s/submission-guidelines#loc-laboratory-protocols

We look forward to receiving your revised manuscript.

Kind regards,

Julien Martin

Academic Editor

PLOS ONE

Additional Editor Comments (if provided):

All minor revision have been addressed correctly, and I would be pleased to accept the manuscript. However, I am asking for 1 minor revision before acceptance.

L. 67-68 "In summary, selection shapes individuals to be an ideal size to maximize reproductive success". I found that sentence awkwardly phrased. I realized it was modified following suggestions from reviewer 1 but the sentence still is not adequate. Since the sentence is neither providing extra information nor improving the flow of the manuscript, I would ask you to simply remove it.

Also before resubmitting the final version have a really careful read and make sure that everything is as you want it for publications since PLOS One is not doing any proof editing before publication. This is essentially your last chance to correct typos or minor problems before publications, remove any hidden text or hidden comments from the word document since they would be visible in the published paper.

---

## [Author Response · Author response to Decision Letter 2]

27 Apr 2020

Dear Dr. Martin,

Thanks again for your efforts in helping us improve this manuscript. We have now deleted the one problem sentence as you suggested. 

We have also carefully gone through the manuscript and we are happy for it to be published as is.

Many thanks.

Kind regards,

Edward Wright

---

## [Editor Report · Decision Letter 3]

1 May 2020

Dominance rank but not body size influences female reproductive success in mountain gorillas

PONE-D-19-28202R3

Dear Dr. Wright,

We are pleased to inform you that your manuscript has been judged scientifically suitable for publication and will be formally accepted for publication once it complies with all outstanding technical requirements.

With kind regards,

Julien Martin

Academic Editor

PLOS ONE

---

## [Editor Report · Acceptance letter]

7 May 2020

PONE-D-19-28202R3 

Dominance rank but not body size influences female reproductive success in mountain gorillas 

Dear Dr. Wright:

I am pleased to inform you that your manuscript has been deemed suitable for publication in PLOS ONE. Congratulations! Your manuscript is now with our production department. 

With kind regards,

on behalf of

Dr. Julien Martin 

Academic Editor

PLOS ONE